# Towards Universal Semantics with Large Language Models

## Abstract

The Natural Semantic Metalanguage (NSM) is a linguistic theory based on a universal set of *semantic primes*: simple, primitive word-meanings that have been shown to exist in most, if not all, languages of the world. According to this framework, any word, regardless of complexity, can be paraphrased using these primes, revealing a clear and universally translatable meaning. These paraphrases, known as explications, can offer valuable applications for many natural language processing (NLP) tasks, but producing them has traditionally been a slow, manual process. In this work, we present the first study of using large language models (LLMs) to generate NSM explications. We introduce automatic evaluation methods, a tailored dataset for training and evaluation, and fine-tuned models for this task. Our 1B and 8B models outperform GPT-4o in producing accurate, cross-translatable explications, marking a significant step toward universal semantic representation with LLMs and opening up new possibilities for applications in semantic analysis, translation, and beyond.

## 1 Introduction

Lexical semantics, the study of word meaning, lies at the center of human language and is vital for nearly all language-based tasks. However, representing word meanings in a way that is precise, unambiguous, and transferable across languages remains a major challenge, since human languages are known to contain unique words and concepts that do not directly translate into others. Even seemingly universal words like "green" and "blue," have been shown to simply not exist in some languages (44). This problem is often compounded by everyday approaches to describing word meanings, such as dictionary-style definitions, which often suffer from circularity or rely on culturally specific terms that may seem intuitive to English speakers but are not universally applicable (42; 3).

Modern large language models (LLMs) are overwhelmingly pretrained on English text, enabling strong general natural language processing (NLP) performance (2; 41). However, the dominance of modern English in LLM training data introduces challenges for tasks in low-resource domains that require precise semantic understanding, such as low-resource translation (40) or legal text interpretation (30; 39), as models may struggle to generalize beyond the language-specific semantic representations learned in their training. In low-resource translation, for example, approaches to address this semantic gap have tried leveraging multilingual data to learn more "universal" semantic representations during training (31). Other methods focus on test-time behavior, such as dynamically accessing external lexical resources, like dictionaries, to enable fine-grained semantic interpretation (15; 33; 16). A semantic framework that could serve as a basis for representing and reasoning about word meaning in a universal, unambiguous, and precise way would offer significant benefits for LLMs for not just low-resource translation, but a wide range of semantically demanding tasks.

The Natural Semantic Metalanguage (NSM) (21) is a linguistic framework that proposes a small set of primitive, universal word-meanings, known as *semantic primes* (Figure 1a). These primes are considered primitive because they represent fundamental semantic units that cannot be defined in simpler terms; their meanings are taken to be self-evident. Substantial empirical evidence shows that equivalent primes exist and are lexicalized (i.e., represented by specific words) in most, if not all, languages (25; 35; 25; 26; 46), enabling translation into any language without loss of meaning. Building on this, words, regardless of their complexity or cultural nuance, can be paraphrased using semantic primes (Figure 1b) to reveal a universally translatable meaning. These paraphrases, called

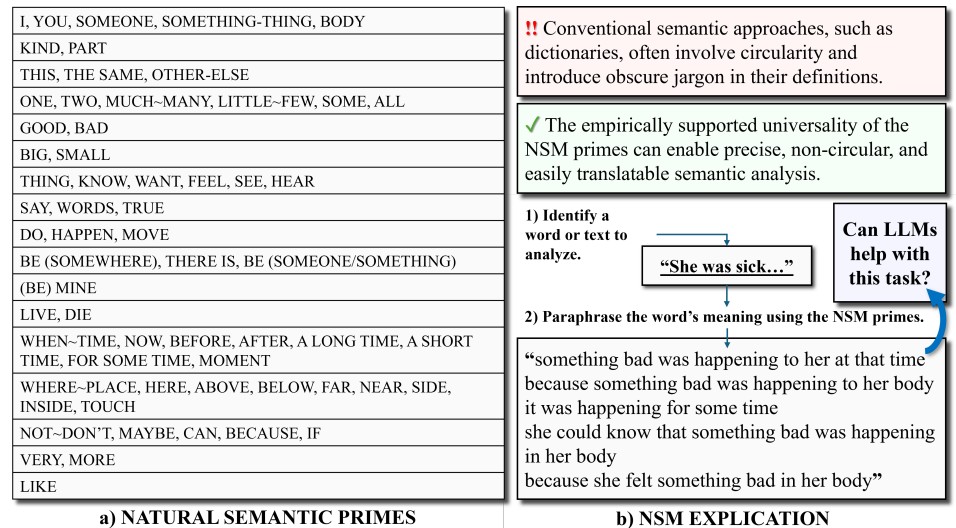

| a) NATURAL SEMANTIC PRIMES | b) NSM EXPLICATION |

I, YOU, SOMEONE, SOMETHING-THING, BODY

KIND, PART

THIS, THE SAME, OTHER-ELSE

ONE, TWO, MUCH~MANY, LITTLE~FEW, SOME, ALL

GOOD, BAD

BIG, SMALL

THING, KNOW, WANT, FEEL, SEE, HEAR

SAY, WORDS, TRUE

DO, HAPPEN, MOVE

BE (SOMEWHERE), THERE IS, BE (SOMEONE/SOMETHING)

(BE) MINE

LIVE, DIE

WHEN~TIME, NOW, BEFORE, AFTER, A LONG TIME, A SHORT TIME, FOR SOME TIME, MOMENT

WHERE~PLACE, HERE, ABOVE, BELOW, FAR, NEAR, SIDE, INSIDE, TOUCH

NOT~DON'T, MAYBE, CAN, BECAUSE, IF

VERY, MORE

LIKE

‼ Conventional semantic approaches, such as dictionaries, often involve circularity and introduce obscure jargon in their definitions.

✓ The empirically supported universality of the NSM primes can enable precise, non-circular, and easily translatable semantic analysis.

1) Identify a word or text to analyze.

"She was sick…"

Can LLMs help with this task?

2) Paraphrase the word's meaning using the NSM primes.

"something bad was happening to her at that time because something bad was happening to her body it was happening for some time she could know that something bad was happening in her body because she felt something bad in her body"

Figure 1: A list of the proposed Natural Semantic Primes, along with an example demonstrating how the NSM framework is applied to paraphrase a word into an explication of its meaning. The list of NSM Primes is adapted from (35). The pictured explication is adapted from (28).

explications, provide a clear, universal way to describe and compare word meanings in ordinary language while avoiding the circularity, jargon, or culture-specific assumptions often found in dictionary definitions or formal semantic systems. These advantages have led to the application of the NSM framework in many fields, including cross-cultural communication (22; 26), cultural analysis (43; 10; 45; 24), language teaching (6; 38), and beyond (29; 11; 50).

If scaled, the NSM framework could offer a foundation for universal semantic representation in LLMs and NLP systems, supporting a range of downstream tasks. However, drafting explications has traditionally been a slow, manual process, with experts sometimes taking weeks or months to produce a single explication (23). Surprisingly, no previous research has explored how machine learning can be used to automate or accelerate this process. In this work, we present the first investigation into using LLMs for universal semantic analysis within the NSM framework, focusing specifically on generating and evaluating NSM explications. We introduce the NSM framework and define a task setup (Section 2), and identify three key challenges in adapting LLMs: the lack of effective models for generating NSM explications, the absence of a suitable dataset or benchmark, and the need for automated methods to evaluate the legality, quality, and cross-translatability of explications. Our contributions are as follows:

1. We propose the first automated methods for evaluating the legality, descriptive accuracy, and cross-translatability of NSM explications (Section 3).

2. We construct the first dataset for training and evaluating LLMs on this task, using our evaluation methods as a quality filter (Section 4).

3. We introduce DeepNSM, two fine-tuned LLMs (1B and 8B), and show that they outperform general LLMs such as GPT-4o and Gemini in explication quality (Section 5).

4. We show that NSM explications have significantly higher cross-translatability into low-resource languages than traditional semantic approaches, such as dictionary definitions, aligned with their proposed universality (Section 5).

The methods, datasets, and models introduced in this work constitute a pivotal first step in adapting LLMs for the NSM framework, establishing a foundation for future research. It opens the door to a universal semantic layer for AI, with far-reaching implications for linguistic study and real-world language applications. All code, models, and datasets introduced in this work are open-sourced.

## 2 Natural Semantic Metalanguage

### 2.1 Overview of the NSM Approach

Figure 1a presents the full list of English exponents of the NSM semantic primes, which have been empirically attested in over 90 languages and counting (25; 35; 25; 26; 46), including many low-resource and endangered languages. The NSM approach is based on the principle that the meaning of any text, regardless of its complexity, can be fully paraphrased using only the semantic primes. This approach can be applied to individual words, multi-word expressions (MWEs), proverbs (27), and longer texts (47).

As illustrated in Figure 1b, the analytical process typically begins by selecting a target word (or span of text) and gathering contextual examples in which the word is used in the same way. Researchers then examine these examples to draft an explication, paraphrasing the word's meaning using only NSM primes. This process is also known as reductive paraphrasing (20; 23), and follows a few core principles. The main guideline is to rely as much as possible on the semantic primes. However, explications, especially in early drafts, do not need to be fully reduced to primes to be useful. Explications are allowed to include non-prime words, referred to as *semantic molecules*, to manage complexity. While some molecules are considered near-universal, many are culture-specific (21). In all cases, the goal is to be as reductive and non-circular as possible. Molecules should only be used when truly necessary, for instance, to simplify a very long explication, should be simpler than the word being paraphrased, and ideally are words that have already been paraphrased in the semantic primes.

Once an explication is drafted, it is tested by substituting it for the target word in the original usage examples to check whether it fully preserves the target word's meaning, captures its key entailments, and remains as minimal as possible (a.k.a. *minimality*). In addition, the explication should be easy to translate across different languages and cultural contexts. Even if written in English, a paraphrase composed largely of semantic primes is less likely to be distorted in translation, as using the primes simplifies the translation task from a semantic challenge to primarily a grammatical and syntactical one.

### 2.2 Applications of NSM

Once a word has been paraphrased using semantic primes, the resulting explication can support a range of downstream applications. A key example is translation, particularly for low-resource languages without direct equivalents for many English terms. For instance, translating "color" into Warlpiri (44), which lacks dedicated color terms, becomes easier when "color" is reduced to a semantically transparent NSM explication, shifting the challenge from semantics to syntax and grammar. Similarly, the universality of semantic primes has led to the application of the NSM framework for language learning and language revival efforts (6; 38). Explications have also been used in literary analysis, such as examining historical texts for semantic drift and cultural changes (45; 10). Automating explication generation could extend this application into traditional NLP tasks like sentiment analysis, document retrieval, or legal text analysis (39; 30), where capturing subtle emotional or contextual meaning is crucial—an area where NSM representations could offer more transparent and precise semantic grounding.

The universality and interpretability of the semantic primes could also provide interesting applications for LLMs themselves. Models that generate text using semantic primes could make the outputs of LLMs and other AI models accessible to speakers of all languages. Explications could support more transparent and interpretable model reasoning by grounding outputs in culturally neutral terms. The NSM framework could even inform LLM architecture design, for example, using semantic primes as a foundation for human-interpretable embedding dimensions, as proposed in recent work (7).

### 2.3 Opportunity of LLM for NSM & Task Setup

While the NSM is an attractive and promising approach to assist NLP/ML tasks in numerous ways, the substantial time required to draft explications has been a critical limitation. Because the process of reductive paraphrasing is not governed by strict formal rules, it resists automation through rule-based systems. As a result, even experienced NSM practitioners may spend weeks or months crafting a

single explication (23). The potential for LLMs to assist in this process presents an opportunity to scale the NSM approach, address one of its key drawbacks, and integrate it into broader NLP systems, such as machine translation pipelines or language learning tools. We define a basic task setup for using LLMs to assist in generating NSM explications. In our formulation, a user selects a target word and provides several contextual examples illustrating its use. The goal is for the model to paraphrase the word's meaning using only the semantic primes. An example of this user-assistant interaction is shown in Figure 8 in the Appendix. In the following section, we examine the challenges of using LLMs to perform this task.

## 2.4 CURRENT CHALLENGES IN USING LLMS FOR THE NSM APPROACH

**Lack of Effective and Reliable Models.** To evaluate current LLMs' ability to generate NSM explications, we construct a system prompt based on the task setup in Section 2.3, using three few-shot examples drawn from freely available expert-authored NSM explications. We apply this prompt to instruction-tuned LLMs of various sizes: Llama-3.2-1B, Llama-3.1-8B, Gemini 2.0-Flash, and GPT-4o. As we show later in Table 1 in Section 5, Smaller models like the Llama variants consistently fail to generate valid explications using the NSM primes, even after prompt tuning. Larger models such as GPT and Gemini produce outputs that more closely follow the instructions and use the semantic primes. Still, their generated paraphrases often fail to capture the target word's meaning accurately. The absence of a model that is both effective and efficient for this task highlights a critical gap in using LLMs to generate NSM explications. Full prompts and sample outputs are provided in the Appendix (Sections D.2, E).

**Lack of Datasets.** Although the absence of effective models creates an opportunity to adapt LLMs for generating NSM explications, progress in that area is prevented by the lack of any datasets for this task. The volume of publicly available, human-authored explications is too limited to support effective fine-tuning, and most existing examples are published under licenses that prevent their inclusion in open datasets. Moreover, these examples would need to be augmented with associated target words and example usages to align with the task setup described in Section 2.3. The situation is further complicated by the absence of any standardized benchmarks or evaluation sets, making it difficult to assess or compare future methods consistently.

**Lack of Methods for Automatically Evaluating Explication Quality.** Critically, the usefulness of any model, dataset, or benchmark is limited without a method for automatically evaluating the quality of generated explications beyond manual analysis. Scalable NSM research requires automated evaluation across three key dimensions: *legality* (correct use of semantic primes and avoidance of circularity), *descriptive accuracy* (how well the explication captures the target word's meaning), and *cross-translatability* (reliable translation across different languages without distortion of meaning). Developing automated methods for evaluating these aspects is essential for advancing this task. The following section presents the first methods for evaluating NSM explications automatically.

## 3 PROPOSED METHODS FOR AUTOMATIC EVALUATION OF NSM EXPLICATIONS

In this section, we introduce three automatic evaluation methods for assessing the legality, descriptive accuracy, and cross-translatability of NSM explications.

### 3.1 EXPLICATION LEGALITY SCORING

We score the legality of an NSM explication based on the ratio of primes to total words and the ratio of semantic molecules. Non-prime words that are not included in the NLTK stopwords list (9) are treated as molecules, because stopwords that are non-prime are typically simple grammar words that carry little or no semantic information. The legality score is then calculated as:

$$\text{Legality Score} = \frac{\alpha * (\text{primes} - \text{molecules})}{\text{total words in explication}} \quad (1)$$

Here, $\alpha$ is a tunable constant weighting the Legality Score in Equation 3. We set $\alpha$ to 10, so the score ranges from -10 to 10, with positive values indicating more semantic primes than molecules in the explication.

## 3.2 EVALUATING DESCRIPTIVE ACCURACY (SUBSTITUTABILITY TEST)

As discussed in Section 2.1, NSM researchers assess an explication's descriptive accuracy by substituting it for the target word, ensuring meaning is preserved, and verifying minimality and correct entailments. Our goal is to develop an automated method based on this practice.

(1) We start with selecting an ambiguous passage $x$ containing the target word $w$, and mask it as <UNK>, ensuring that the target word cannot be easily predicted without additional clues. We then prompt an LLM to predict the masked word and measure the log-probability of the target word's token(s). Next, we repeat the process, this time providing the explication $e$ for the masked target word (simulating its substitution) and prompting the LLM to predict the masked word. We measure the change in log-probability, defined as: $\Delta_{\textbf{baseline}} = \log p(w|x, e) - \log p(w|x)$. An effective explication should increase the LLM's likelihood of predicting the correct word compared to when no explication is given, and should yield a larger positive delta compared to a poor explication.

(2) To assess the minimality of an explication, we sequentially remove $k$ lines, one at a time, from the end of the explication, and calculate the average change in log-probability over both removals, defined as: $\Delta_{\textbf{min}} = \sum_{i=1}^{k} \log p(w|x, e_{-i}) - \log p(w|x, e_{-i+1})$. Here, $e_{-i+1}$ refers to the explication with 1 less line removed, or the full explication if $i = 1$. We set $k = 2$, given that explications will almost always contain 3 or more lines. If an explication is redundant or contains unnecessary details, this change should be small or even positive as lines are removed. In contrast, an appropriately minimal explication should show a negative delta, indicating a loss of meaningful information in the explication with each removal.

(3) To test whether an explication captures the target word's entailments, we sequentially remove $k$ sentences, one at a time, from the passage (excluding the one with <UNK>) and measure the average change in log-probability, defined as: $\Delta_{\textbf{ent}} = \sum_{j=1}^{k} \log p(w|x_{-j}, e) - \log p(w|x_{-j+1}, e)$. We use the same $k = 2$ from the minimality tests for consistency. Here, $x_{-j+1}$ refers to the passage with 1 less line removed, or the full passage if $i = 1$. If the model's prediction remains stable or improves, it suggests the explication contains the core meaning needed to infer the word. A drop in probability implies the model relied on information from the removed context—entailments the explication failed to supply. For example, if removing "She was feeling very sad" from the passage lowers the model's confidence in predicting "cry," the explication likely missed conveying that emotional state.

For an explication, we perform these tests repeatedly over multiple ambiguous passages $P$ and LLMs $G$, averaging the outcomes from all of them in order to generalize the result and reduce sensitivity to any particular passage or model behavior. The LLMs we use in this process include instruction-tuned Llama-3.1-8B, Mistral-7B, and Gemini-3-12B. Finally, we calculate a "substitutability score" to assess the descriptive accuracy of an NSM explication, with the following formula:

$$\text{Substitutability Score} = \frac{1}{|G||P|} \sum_{g \in G} \sum_{p \in P} \left( \min(\beta, \Delta_{\text{baseline}}^{(g,p)} - \Delta_{\text{min}}^{(g,p)} + \Delta_{\text{ent}}^{(g,p)}) \right) \quad (2)$$

This scoring formula rewards explications that are descriptively accurate (improving target word prediction), minimal, and have the right entailments. $\beta$ is a cap that can be set to limit the maximum substitutability score, preventing extreme values from skewing the evaluation. We set $\beta$ to 40, as we observed that for all LLMs tested, log-probability gains beyond this point were very rare but could be extremely large; this also helps keep the overall explication score (Section 3.3) within a 100-point scale. To our knowledge, this is the first automated method for evaluating NSM explication accuracy without requiring human input. We include pseudocode for this process in Section B.2.

## 3.3 OVERALL EXPLICATION SCORE

To provide a metric that can holistically assess both the legality and descriptive accuracy of an explication, we compute an overall explication score by combining the above two metrics:

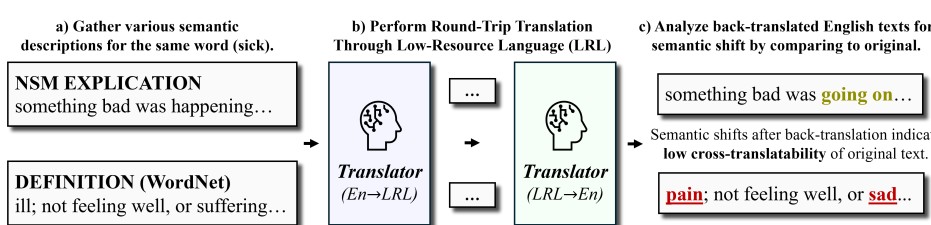

Figure 2: Illustration of the cross-translatability test described in Section 3.4. Semantic descriptions are translated into a low-resource language supported by Google Translate and then back into English. Any semantic drift in the back-translated text indicates difficulty in consistent translation.

$$\text{Explication Score} = \gamma * (\text{Substitutability Score} + \text{Legality Score}) \qquad (3)$$

We set $\gamma$, a tuneable constant, to 2 to normalize the max score to 100, with the substitutability score contributing up to 40 points and the legality score ranging from -10 to 10. Circular explications (containing the target word) are automatically scored as 0. Future work may explore alternative settings of $\alpha, \beta, k$, and $\gamma$.

### 3.4 Evaluating Cross-Translatability

A key advantage of NSM explications is their cross-translatability. To test the cross-translatability of an explication (or any text), we propose the procedure illustrated in Figure 2. We collect various semantic descriptions for a word, such as NSM explications and dictionary definitions. Using Google Translate as our translator, we focus on low-resource languages with known lower translation quality. The semantic descriptions in English text are translated into the target language and then back into English. We assess back-translated English texts by comparing them to their originals, using BLEU scores and embedding-based similarity to measure lexical and semantic differences. Lower scores on either metric indicate greater semantic drift, suggesting more difficulty in consistent translation in and out of the target language. Although NSM explications are longer than dictionary definitions, they are expected to perform better due to their use of semantic primes, which should reduce the semantic complexity of the translation task.

## 4 Proposed Dataset

In this section, we introduce the first dataset designed explicitly for the task of NSM explication generation, which serves as the basis for fine-tuning the DeepNSM model described in Section 5. Following the task formulation outlined in Section 2.3, each entry in the dataset includes a target word, several example sentences illustrating its usage, and a corresponding NSM explication.

**Dataset Generation.** To ensure our dataset covers a wide range of word-meanings, we use WordNet (34), a lexical database that provides structured definitions for various word senses, grouped into synsets. Synsets are sets of synonymous words that share a single sense or meaning. After filtering out NSM primes, we obtain 88,078 unique word senses. While WordNet includes some example sentences, many synsets lack examples, and none offer ambiguous passages which could be used for the substitutability tests we define in Section 3.2. Figure 3 demonstrates our dataset generation process. We first select a target word sense from a WordNet synset, applying filters to exclude NSM primes. To address gaps in example sentences, we use instruction-tuned LLMs (Llama-3-1/8B, Mistral-7B, Gemma-2-2/9B, Llama-3.2-3B, Gemma-2-3B) to generate additional examples, prompted with the WordNet definition to ensure the examples align with the word sense.

For each word sense, we gather 2 to 5 example sentences using the target word in context. We then prompt an LLM to generate multiple candidate NSM explications based on the target word and its examples. We select Gemini-2.0-Flash for this step, as it offers a good balance between explication quality and speed. With a temperature setting of 0.7, Gemini generates over 440,000 candidate explications across all synsets. As noted in Section 2.4, the output quality of general-purpose LLMs

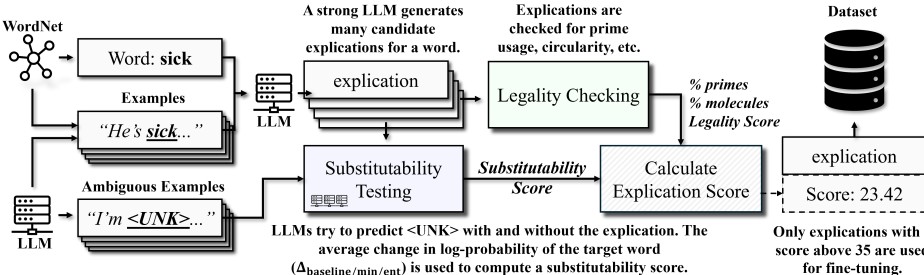

Figure 3: Our dataset generation process described in Section 4. Target words are selected from WordNet and paired with example usage sentences. Gemini-2.0-Flash generates many candidate explications for the word, and the evaluation methods outlined in Section 3 are used to select the highest quality explications for the final dataset.

can vary significantly. To address this, we use the evaluation metrics from Section 3 to identify and remove low-quality generations. The filtered dataset aims to support fine-tuning more effective and consistent LLMs for NSM explication generation.

Once candidate explications are generated, we first perform legality checks on each and calculate a legality score (Equation 1). Then, we assess descriptive accuracy through substitutability testing, generating four ambiguous example paragraphs with an LLM. We compute a substitutability score for each explication using three grader LLMs (Llama-3.1-8B, Mistral-7B-v0.3, and Gemma-3-12B), following Section 3.2. Legality and substitutability scores are then combined to compute the overall Explication Score (Equation 3). This score serves as a quality indicator that can be used to filter out low-quality entries from the final dataset. Finally, the word, scored explication, and example sentences are added as a candidate entry in the dataset.

**Evaluation Set/Benchmark.** For the evaluation split, we hand-curated 149 unique words across various semantic categories, such as nouns, mental predicates, and speech act verbs. Each word was mapped to its WordNet synsets to ensure that neither the word, its synonyms, nor alternative senses from the training data were included, preventing contamination. We selected two to five usage examples and four ambiguous paragraphs for consistent substitutability testing, with examples and contexts manually refined for clarity and coherence.

**Quality Filtering and Final Dataset.** After generating and scoring candidate explications, we filter out those with scores below 35, a quality threshold met by fewer than 15% of candidates, as most scores range from 10 to 20. To prevent overrepresentation, we cap explications per word sense at two. The final dataset contains around 44,000 word-example-explication entries, with 1,000 set aside for validation and 43,000 for training. The hand-curated test set of 149 entries is used for evaluation. We plan to release this dataset upon publication.

## 5   DEEPNSM MODEL AND EXPERIMENTS

In this section, we present DeepNSM, the first LLM fine-tuned specifically for generating NSM explications. We evaluate DeepNSM against several baseline LLMs and demonstrate that fine-tuning on our dataset introduced in Section 4 enables DeepNSM to produce NSM explications that are both higher in quality and more easily translatable across languages.

**Setup.** To address key model limitations described in Section 2.4, we fine-tune Llama 3.2 1B and Llama 3.1 8B models using our curated dataset. All fine-tuning runs are conducted for one epoch on an NVIDIA H100 GPU. We evaluate all models using the evaluation set described in Section 4. For each target word in the benchmark, models are prompted to generate an explication given the word and its examples. All baseline models are prompted with a consistent system message and a fixed set of few-shot examples. The generated explications are evaluated using the legality, substitutability, and cross-translatability tests described in Section 3, as well as qualitative human ranking. We compare DeepNSM models (1B and 8B) with the following instruction-tuned LLM baselines: Llama-3.2-1B, Llama-3.1-8B, Gemini-2.0-Flash (estimated ∼70B parameters), and GPT-4o (200B estimated (1)).

Table 1: Evaluation of NSM explications generated by LLMs for the benchmark set introduced in Section 4. An up (down) arrow for a metric means higher (lower) is better. "Dictionary Def." are existing definitions provided from WordNet. Best underlined. † means trained on a dataset with no quality filtering applied. DeepNSM models surpass SOTA general LLMs for NSM explication generation despite only having 1B and 8B parameters.

| Model | Explication Score ↑ | Legality Score ↑ | Substitutability Score ↑ | Primes Ratio ↑ | Molecules Ratio ↓ | Circular % ↓ |
|---|---|---|---|---|---|---|
| Dictionary Defs. | **13.4** | -4.7 | 12.14 | 8.0 | 55.1 | 10.0 |
| Llama-3.2-1B-it | **6.1** | -0.4 | 6.82 | 29.2 | 33.2 | 45.6 |
| Llama-3.1-8B-it | **20.0** | 2.3 | 7.86 | 43.9 | 20.9 | 2.0 |
| Gemini-2.0-Flash | **22.2** | 5.1 | 6.40 | 62.1 | 11.5 | 2.7 |
| GPT-4o | **22.9** | 4.6 | 6.95 | 59.9 | 13.8 | 1.3 |
| DeepNSM-1B† | **19.7** | 5.0 | 5.22 | 61.1 | 10.9 | 4.7 |
| DeepNSM-8B† | **22.2** | 4.9 | 6.40 | 60.0 | 11.5 | 1.3 |
| DeepNSM-1B | **23.2** | 5.1 | 7.02 | 61.9 | 10.6 | 5.4 |
| DeepNSM-8B | **24.6** | 5.4 | 7.34 | 63.9 | 10.4 | 3.3 |

We also include the existing WordNet definitions ("Dictionary Def.") as a non-LLM, non-NSM baseline. Similar to fine-tuning, all evaluations are conducted on an NVIDIA H100 GPU. Error bars for all experiments can be viewed in Tables 3, 4, and 5 in the Appendix.

**DeepNSM Enables High-Quality, Efficient NSM Explication Generation.** Table 1 shows that despite having only 1B and 8B parameters, all DeepNSM variants consistently outperform or match state-of-the-art general-purpose models across key metrics, including legality, substitutability, prime usage, molecule usage, and circularity. Notably, DeepNSM models achieve the highest overall explication score, with even the 1B model outperforming all other models, regardless of size. While dictionary definitions and Llama baselines score higher on substitutability, these results are misleading because they rely on poor prime usage and high circularity, often reusing the target word, effectively "cheating" the NSM paraphrasing task. In contrast, the fine-tuned DeepNSM 1B and 8B models improve prime usage by 30.9% and 16.1% and reduce molecule usage by 22.3% and 9.4%, respectively. These findings demonstrate that high-quality NSM explication generation is achievable in smaller models with our proposed dataset, addressing the model limitations discussed in Section 2.4 without requiring large-scale compute or commercial APIs.

**Dataset Quality Filtering Improves Explication Accuracy and Prime Usage.** To isolate the impact of dataset quality filtering, we fine-tune two versions of each DeepNSM model: one on the high-quality, filtered dataset (Explication Score $\geq 35$) described in Section 4, and another on a randomly sampled, unfiltered dataset of the same size (marked with †), drawn from the full pool of candidate explications without regard to quality. Results in Table 1 show that, compared to their unfiltered counterparts, the DeepNSM 1B and 8B models trained on quality-filtered data achieve a 34% and 15% relative increase in substitutability scores, respectively. Prime usage also improves, with the DeepNSM 8B model showing a 3.9 percentage point gain. These improvements demonstrate the effectiveness of our filtering strategy in enhancing both descriptive accuracy and semantic prime usage. By producing the first high-quality dataset for NSM explication generation, our work directly addresses the dataset limitations outlined in Section 2.4.

**NSM Explications Demonstrate High Cross-Translatability for Low-Resource Languages.** We perform cross-translatability testing (Section 3.4) on model-generated explications with five low-resource languages: Alur, Dinka, Kinyarwanda, Dzongkha, and Abkhaz. We then measure the match between the back-translated explication and the original using BLEU and embedding similarity, with embeddings generated from BERT-based sentence embeddings in the Sentence-Transformers library (37). Table 2 shows that NSM explications are significantly more resistant to semantic drift during translation to and from low-resource languages compared to dictionary-style definitions. This is particularly evident in Kinyarwanda, where NSM explications outperform by over 20 BLEU points and 10 embedding similarity points. DeepNSM-generated explications also achieve the highest cross-translatability for languages Alur, Dinka, Dzongkha, and Abkhaz. These results suggest that

Table 2: Cross-translatability results following the method described in 3.4. We test on model-generated explications for five low-resource languages from different families: Alur (alz), Kinyarwanda (rw), Dzongkha (dn), Dinka (din), and Abkhaz (ab). After performing round-trip translation, we measure the semantic shift of the back-translated text using BLEU and Embedding Similarity. Best underlined. NSM explications consistently show less semantic drift, reflecting their universality.

| Model | BLEU ↑ | | | | | Embedding Similarity ↑ | | | | |
|---|---|---|---|---|---|---|---|---|---|---|
| | alz | rw | dn | din | ab | alz | rw | dz | din | ab |
| Dictionary Def. | 22.9 | 35.1 | 19.6 | 24.4 | 26.9 | 69.6 | 79.6 | 73.3 | 69.1 | 78.0 |
| Llama-3.2-1B-it | 29.0 | 45.0 | 21.0 | 25.8 | 25.2 | 79.3 | 88.0 | 81.3 | 77.3 | 84.2 |
| Llama-3.1-8B-it | 33.2 | 50.8 | 23.8 | 31.8 | 27.8 | 81.6 | 89.8 | 81.8 | 78.1 | 84.0 |
| Gemini-2.0-Flash | 33.5 | 56.3 | 23.2 | 32.2 | 30.1 | 84.0 | 93.0 | 81.7 | 78.5 | 85.1 |
| GPT-4o | 31.2 | 51.3 | 21.6 | 31.2 | 28.7 | 82.0 | 91.4 | 78.9 | 77.4 | 84.8 |
| DeepNSM-1B | 34.5 | 56.9 | 23.6 | 32.6 | 31.9 | 81.6 | 91.4 | 80.4 | 78.0 | 82.7 |
| DeepNSM-8B | 37.0 | 55.6 | 23.3 | 32.4 | 33.2 | 84.2 | 91.7 | 82.0 | 79.0 | 85.4 |

DeepNSM could already serve as a tool for helping rephrase English texts to make them more accessible and readily translatable for speakers of many languages.

**Metrics Align with Qualitative Judgements.** We also conduct a qualitative analysis to evaluate the alignment between our automatic metrics and human judgments. To do this, we perform a blind review and ranking of the explications generated by all models evaluated in Table 1. Explications from DeepNSM models received top rankings 46% of the time, with 33% for the 8B model and 13% for the 1B model, which frequently placed second to 8B. This was significantly higher than the top rankings received by GPT (28%), Gemini (21%), and the Llama models (5%). These findings indicate that automatic metrics align with human qualitative analysis and highlight the effectiveness of the dataset filtering strategy, as DeepNSM trained on filtered data consistently outperforms other models. These results support the effectiveness of our metrics in addressing the critical challenge outlined in Section 2.4, providing the first automated evaluation method for NSM explications.

## 6 RELATED WORK

Prior work has applied NSM to computational tasks, such as encoding NSM texts in PROLOG (49), creating word vectors aligned with semantic primes (7), and using NSM for building semantic graphs (17). However, these methods do not use deep learning or target LLMs, and none address automatic explication generation, which is the central focus of our work. (18) use LLMs to study semantic drift by generating contextualized definitions from target words and usage examples. However, their work does not target NSM, and thus these definitions can be inherently more prone to circularity and poor cross-translatability. Other work (36) tracks semantic drift through changes in LLM-derived embeddings. Incorporating NSM into such analyses is an promising research direction which can be catalyzed by this work. Multilingual embeddings (12; 13) project semantically related words from different languages into a shared space to support cross-lingual tasks. Yet, they often lack the semantic precision of English-focused LLMs and depend heavily on large multilingual corpora. (31) align a multilingual encoder with an English-centric LLM to create a "universal embedding space." However, this space is purely learned and lacks grounding in any semantic framework. Such work highlights broader interest for universal semantic representations in LLMs and this work's relevance.

## 7 CONCLUSION

This work introduces the first use of LLMs for the NSM approach to semantic analysis, along with the first models, dataset, and evaluation methods, laying the groundwork for future research. The potential impact is significant, as we believe AI can scale the NSM approach to reveal entire languages as vast articulations of semantic primes, enabling new possibilities for cross-linguistic and cross-cultural understanding, as well as promising applications for many tasks. Readers may see the Appendix for further details and discussion of all content presented in the paper.

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

# Appendix

This appendix offers supplementary materials, background information, and extended discussions that could not be included in the main paper due to space constraints.

## A Background on Natural Semantic Metalanguage

### A.1 Semantic Primes

Semantic primes are treated as irreducible semantic units that cannot be defined using simpler terms without leading to circularity, where either the word itself appears in its own definition, or the words used to define it lead back to the original word. These meanings are taken to be self-evident and universally present across languages. A particular language's translation of the semantic primes are often referred to as exponents. Importantly, primes can display polysemy in specific languages, potentially leading to misinterpretation (26; 21; 25). For instance, the English exponent "above" is a valid prime in the spatial sense ("the sky is above the ground"), but not in the metaphorical sense ("I am above this work"). Additionally, primes may exhibit allolexy, that is, they may be represented by multiple synonymous words in a language (21; 25; 26). Semantic primes have also been proposed to have associated grammatical properties (21).

### A.2 Methodological Advantages of NSM

The NSM framework is built on the view that natural languages are inherently capable of representing their own semantics through internal paraphrase, without the need for an external, formalized metalanguage. While formal semanticists and logicians may see natural language as too imprecise for semantic representation, as (5) argues, even trained experts inevitably rely on their native languages when learning formal systems and when discussing the intuitions behind their formalisms. Furthermore, to support testable predictions about real language use, a semantic description requires a clear and transparent link between the semantic description and the natural language it aims to explain. When that link becomes too abstract or detached, empirical validation can become more challenging. For instance, the common formal analysis of X kills Y as [CAUSE x (DIE y)] fails to account for usage differences between "kill" and "cause to die," yet rather than revising the analysis, some have argued that CAUSE doesn't mean the same as the English word cause (32). However, such a move can undermine empirical testing by insulating the analysis from being supported by real-world evidence.

Another methodological benefit of restricting semantic analysis to a set of universal semantic primes is that it helps avoid presumptively imposing foreign cultural or terminological categories onto the languages being studied. This is especially important when working with less widely spoken or native languages, where introducing a formal metalanguage may impose a semantic component CAUSE onto a language where the word cause has no corresponding verb (19). Such impositions risk misrepresenting the language's conceptual distinctions and culturally specific meanings. By relying solely on universal semantic primes for analysis, the NSM approach ensures that semantic descriptions remain free from any culturally specific vocabulary and thus broadly cross-translatable.

Moreover, NSM paraphrases still preserve all the advantages of a highly disciplined system for semantic analysis. By limiting paraphrases to semantic primes, the framework enables direct and structured comparisons between word meanings, helping to reveal subtle differences or shared components that might be difficult to detect in natural language alone. For example, we can directly compare the words ill and sick via their explications, taken from (28):

[A] *she was ill*

**a** something bad was happening to her at that time

   because something bad was happening in her body

**b** it was happening for some time, not a short time

**c** she felt something bad at that time, not like at other times

   she couldn't do some things at that time like at other times

[C] *she was sick*

**a** something bad was happening to her at that time

  because something bad was happening in her body

**b** it was happening for some time

**c** she could know that something bad was happening in her body

  because she felt something bad in her body

A close comparison of these explications reveals a key distinction: in line b, ill includes the additional phrase "not a short time." This suggests that ill is typically used to describe longer-term conditions, while sick may apply to shorter or even momentary states of discomfort. From this difference, we can formulate testable predictions about how these words are used. For instance, if ill tends to describe longer durations, we would expect it to occur less frequently in contexts with expressions like briefly, momentary, or even the semantic prime a short time. If explications are restricted to semantic primes, then generating and testing these kinds of predictions about real-world language use, such as differences in word usage or co-occurrence patterns, can be systematically automated, i.e., testing for the aforementioned collocations whenever "not a short time" appears in an explication.

# B  FURTHER DETAILS & JUSTIFICATIONS FOR PROPOSED EVALUATION METHODS

Here, we expand on the evaluation methods introduced in Section 3, offering detailed motivation, descriptions, and justifications where appropriate. We also include examples and figures to illustrate key points where helpful.

## B.1  EXPLICATION LEGALITY

To evaluate an explication, the first, most basic step is to assess how well it follows the principles of the NSM framework: it should rely primarily on semantic primes, minimize the use of semantic molecules, and avoid circularity. This requires checking how extensively the explication uses semantic primes, how many molecules it includes, and whether it contains the original word it is meant to define. To implement this, we parse the explication and remove all punctuation. We then count the number of words that are semantic primes. Any remaining words that are neither semantic primes nor members of the NLTK stopwords list (9) are counted as molecules. We exclude stopwords from being counted as molecules because we observe most serve grammatical functions and contribute little semantic information. To detect circularity, we check if any form of the original word is contained in the explication.

Next, we define a metric to quantify how "legal" an explication is; that is, how closely it adheres to the NSM framework. We propose the following formula:

$$\text{Legality Score} = \frac{\alpha * \text{primes} - \text{molecules}}{\text{total words in explication}} \qquad (4)$$

Here, $\alpha$ is a tunable constant weighting the Legality Score in Equation 3. We set $\alpha$ to 10, so the score ranges from -10 to 10, with positive values indicating more semantic primes than molecules in the explication. A score of 10 represents a "perfectly legal" explication composed entirely of semantic primes, while a score of -10 indicates that no semantic primes are used. In general, this formula will reward explications that use a higher ratio of primes than molecules. We base the calculation on word ratios to normalize for differences in explication length. Although this metric does not account for circularity, we address that separately in the Explication Score metric defined in Section B.4.

## B.2  DESCRIPTIVE ACCURACY (SUBSTITUTABILITY)

While the Legality Score captures how many semantic primes and molecules are used in an explication, it does not assess how accurately the explication describes the meaning of the target word. As discussed in Section 2.1, NSM researchers typically evaluate this by substituting an explication with

the word in various usage examples and checking whether the meaning is preserved. They also ensure that the explication is minimal, avoiding unnecessary or redundant information, and that it includes all the correct semantic entailments. Our goal is to develop an automated method for performing this type of substitutability testing.

In order to make our evaluation possible, we need to collect passages that contain the target word, which we can use for substitution testing. We use three-sentence passages in which exactly one sentence contains the target word, masked as <UNK>. The core idea behind our method is to use LLMs to try to guess the <UNK> word, both with and without access to the word's explication. However, LLMs can often guess the missing word easily, even without an explication, especially if the surrounding context is highly informative. For example, in the sentence *"I loved basketball as a kid, so I was extra excited when my dad got me a basketball <UNK> so I could practice shooting at home,"* it is easy to infer that <UNK> is "hoop," regardless of whether an explication for "hoop" is provided. Because of this, we elect to prompt GPT-4o mini to generate intentionally ambiguous example passages, making the masked word less obvious for an LLM to guess without additional clues. We provide GPT-4o mini with the target word and a definition of its intended sense, and ask it to generate three-sentence paragraphs where the word is used naturally and masked as <UNK>. Figure 10 shows an example of this prompt.

Once we have an ambiguous passage $x$ with the target word $w$ masked as <UNK>, we prompt an LLM, referred to as the grader LLM, to predict the masked word by measuring the probability it assigns to the correct token(s). We first run this prediction using only the passage. Then, we repeat the process, but this time we also provide the grader with the explication of the masked word. This setup simulates replacing the target word with its explication. If the explication accurately captures the word's meaning, it should help the grader assign a higher probability to the correct word. We quantify this effect by measuring the change in log-probability of the correct word before and after the explication is given, defined as: $\Delta_{\textbf{baseline}} = \log p(w|x, e) - \log p(w|x)$. An explication that accurately describes the word's meaning should yield a positive delta, larger than one that fails to do so.

To assess the minimality of an explication, we sequentially remove $k$ lines, one at a time, from the end of the explication, and calculate the average change in log-probability over both removals, defined as: $\Delta_{\textbf{min}} = \sum_{i=1}^{k} \log p(w|x, e_{-i}) - \log p(w|x, e_{-i+1})$. Here, $e_{-i+1}$ refers to the explication with 1 less line removed, or the full explication if $i = 1$. We set $k = 2$, given that explications will almost always contain 3 or more lines. This number reflects how much meaningful information is lost with each line removed from the explication. If the explication includes unnecessary or redundant information, the score will be close to zero or even positive, indicating that the removed lines did not help, or may have even hurt, the prediction. In contrast, a more minimal explication will see a noticeable drop in log-probability with each removal, resulting in a negative score.

To test whether an explication captures the target word's entailments, we sequentially remove $k$ sentences, one at a time, from the passage (excluding the one with <UNK>) and measure the average change in log-probability, defined as: $\Delta_{\textbf{ent}} = \sum_{j=1}^{k} \log p(w|x_{-j}, e) - \log p(w|x_{-j+1}, e)$. We use the same $k = 2$ from the minimality tests for consistency. Here, $x_{-j+1}$ refers to the passage with 1 less line removed, or the full passage if $i = 1$. A positive or unchanged score suggests that the explication provides the essential entailments needed to infer the word, even as context is stripped away. In contrast, a negative score indicates that the model depended on contextual clues from the removed lines that the explication failed to capture.

For an explication, we perform these tests repeatedly over multiple ambiguous passages $P$ and LLMs $G$, averaging the outcomes from all of them in order to generalize the result and reduce sensitivity to any particular passage or model behavior. The LLMs we use in this process include instruction-tuned Llama-3.1-8B, Mistral-7B, and Gemini-3-12B. Finally, we calculate a "substitutability score" to assess the descriptive accuracy of an NSM explication, with the following formula:

$$\text{Substitutability Score} = \frac{1}{|G||P|} \sum_{g \in G} \sum_{p \in P} \left( \min(\beta, \Delta_{\text{baseline}}^{(g,p)} - \Delta_{\text{min}}^{(g,p)} + \Delta_{\text{ent}}^{(g,p)}) \right) \qquad (5)$$

$\beta$ is a cap that can be set to limit the maximum substitutability score, preventing extreme values from skewing the evaluation. This metric rewards explications that make it easier for the grader



---

**Algorithm 1** Compute Substitutability Score (Generalized for any $k$)

---

**Require:** Target word $w$, sense definition $s$, explication $e$, grader LLMs $\{G_i\}$, truncation depth $k$, cap $\beta$

1: Generate ambiguous passages $\{P_j\}$ using GPT-4o mini with $w$ and $s$, masking $w$ as <UNK>
2: **for** each grader LLM $G_i$ **do**
3:     **for** each passage $P_j$ **do**
4:         $x_0 \leftarrow P_j$ with <UNK> masking $w$
5:         $\Delta_{\text{baseline}}^{(i,j)} \leftarrow \log p(w \mid x_0, e) - \log p(w \mid x_0)$
                                                                            ▷ Compute $\Delta_{\min}$
6:         Initialize $\Delta_{\min}^{(i,j)} \leftarrow 0$
7:         **for** $m = 1$ to $k$ **do**
8:             $e_{-m+1} \leftarrow$ explication with $(m-1)$ lines removed from the end
9:             $e_{-m} \leftarrow$ explication with $m$ lines removed from the end
10:            $\Delta_{\min}^{(i,j)} \mathrel{+}= \log p(w \mid x_0, e_{-m}) - \log p(w \mid x_0, e_{-m+1})$
11:         **end for**
12:         $\Delta_{\min}^{(i,j)} \leftarrow \Delta_{\min}^{(i,j)}/k$
                                                                           ▷ Compute $\Delta_{\text{ent}}$
13:         Initialize $\Delta_{\text{ent}}^{(i,j)} \leftarrow 0$
14:         **for** $m = 1$ to $k$ **do**
15:             $x_{-m+1} \leftarrow$ passage with $(m-1)$ non-<UNK> lines removed (excluding the <UNK> line)
16:             $x_{-m} \leftarrow$ passage with $m$ non-<UNK> lines removed (excluding the <UNK> line)
17:            $\Delta_{\text{ent}}^{(i,j)} \mathrel{+}= \log p(w \mid x_{-m}, e) - \log p(w \mid x_{-m+1}, e)$
18:         **end for**
19:         $\Delta_{\text{ent}}^{(i,j)} \leftarrow \Delta_{\text{ent}}^{(i,j)}/k$
20:         $\text{score}_{ij} \leftarrow \min(\beta, \Delta_{\text{baseline}}^{(i,j)} - \Delta_{\min}^{(i,j)} + \Delta_{\text{ent}}^{(i,j)})$
21:     **end for**
22:     $\text{score}_i \leftarrow \text{mean}_j(\text{score}_{ij})$
23: **end for**
24: **return** Average substitutability score: $\frac{1}{|G|} \sum_i \text{score}_i$

---

LLM to predict the target word ($\Delta_{\text{baseline}}$ positive). Explications that are not minimal (i.e., $\Delta_{\text{min}}$ 0 or positive) are penalized, while those that do not capture the correct entailments ($\Delta_{\text{ent}}$ negative) are penalized. Before running evaluations, we verified that these LLMs used for the subsitutability testing produce log-probability values on a comparable scale, as shown in Figure 13. We observed that the models, particularly Gemma, can sometimes assign unusually large log-probabilities, as shown in the figure. A manual review of these instances revealed that these extreme values were likely due to idiosyncrasies, rather than indicative of a particularly accurate explication. As a result, we decided to set $\beta$ to 40. We leave further exploration of alternative weightings for $\alpha, \beta, k$, and $\gamma$ to future work.

### B.3  CROSS-TRANSLATABILITY

A core requirement of NSM explications is that they be easily cross-translatable. To evaluate this property and to highlight the advantages NSM explications may have over traditional semantic descriptions like dictionary definitions, we introduce a method to estimate how well a text can be translated across languages. Since we lack reference translations for most texts, we use round-trip translation as a proxy for cross-translatability. Our approach is shown in Figure 2. We begin with the semantic descriptions to be tested, such as an explication or dictionary definition. Using a machine translation system, we translate the English text into a target language and then back into English. In this work, we use Google Translate and focus on low-resource languages where translation quality tends to be lower to better stress-test the text. To assess how well the meaning is preserved, we compare the original and back-translated English versions using BLEU scores and embedding-based similarity. Lower scores suggest more semantic drift, meaning the text is harder to translate consistently in and out of the low-resource language. Although NSM explications are typically longer than dictionary definitions, their use of semantic primes is expected to make them more semantically stable and easier to translate accurately, as it reduces the semantic complexity of the translation task.

### B.4  OVERALL EXPLICATION SCORE

Now that we have metrics for an explication's legality and descriptive accuracy, we define a single combined score that incorporates both aspects into one comprehensive measure.

$$\text{Explication Score} = \gamma * (\text{Substitutability Score} + \text{Legality Score}) \tag{6}$$

Explications that include the target word are considered circular and automatically receive a score of 0. The total score combines two components: the substitutability score, which can contribute up to 40 points, and the legality score, which ranges from -10 to 10. The substitutability score, with a max of 40, is weighted more heavily to reflect the primary focus on meaning preservation. We set $\gamma$ to 2 to normalize the max score to 100. This factor is arbitrary and may be revised in future work that considers other weighting strategies.

## C  FURTHER DETAILS FOR DATASET

As described in Section 2.4, there are no publicly available datasets of NSM explications, and too few human-written examples to build one from existing sources. To address this, we construct a dataset using large language models and existing lexical resources, applying the evaluation methods from Section 3 to ensure quality. We fully detail the dataset construction process below.

### C.1  DATASET STRUCTURE

Following the task setup in Section 2.3, each dataset entry consists of a target word, two to five example sentences showing how the word is used, and an associated NSM explication for that word.

### C.2  DATASET SOURCE

To ensure our dataset captures a broad range of word meanings, we use WordNet (34), a lexical database that organizes words into synsets, groups of synonyms that share a common meaning.

WordNet also provides a dictionary-style definition for each word sense and, in some cases, includes example sentences demonstrating its usage, though these examples are relatively uncommon. WordNet is particularly useful for capturing polysemy, since different words can belong to the same synset. To build our base vocabulary for dataset generation, we iterate over all individual words in WordNet, rather than just synsets. This allows us to explicitly account for polysemous words with multiple senses. We filter out NSM primes during this process. After filtering, we are left with 88,078 unique word senses.

## C.3    EXAMPLE GENERATION

As noted in Section 4, most words in WordNet either lack example sentences or include only one, while our goal is to provide 2–5 examples per word. To address this, we use instruction-tuned LLMs (Llama-3-1/8B, Mistral-7B, Gemma-2-2/9B, Llama-3.2-3B, and Gemma-2-3B) to generate additional example sentences. Each model is prompted with the WordNet definition to ensure the generated examples match the intended word sense. We apply this process to all 88,078 filtered word senses from WordNet. After generation, each word sense is associated with over 20 example sentences, from which we randomly select 2-5 when constructing the dataset.

WordNet also does not provide ambiguous example passages that can be used for the substitutability testing described in Section 3.2. To address this, we use GPT-4o mini to generate four ambiguous example paragraphs for each target word, formatted specifically to support the substitutability evaluation procedure. Figure 9 and Figure 10 show the prompts used for example generation.

## D    DEEPNSM MODEL AND EXPERIMENTS

We include the experimental results from the main paper with error bars in Table 3, Table 4, and Table 5. Circularity % from Table 1 is not included, as the error was too insignificant.

## D.1    FINE-TUNING DETAILS

We fine-tuned two models—Llama-3.1-8B and Llama-3.2-1B—on the dataset from Section 4 using parameter-efficient fine-tuning with LoRA (48). Both models were trained with the same set of hyperparameters on an NVIDIA H100 GPU. For both models, we used LoRA with a rank of 64, LoRA alpha set to 16, and a dropout rate of 0.1. We enabled 4-bit quantization using the BitsAndBytes library (14) with the nf4 quantization type and bfloat16 compute data type. The optimizer was paged_adamw_32bit with a learning rate of 2e-4, using an inverse square root learning rate schedule and a warmup ratio of 0.03. We used a batch size of 8 with gradient accumulation set to 1 and applied gradient clipping with a maximum norm of 0.3. The sequence length was limited to 512 tokens.

## D.2    EXAMPLE MODEL OUTPUTS

In this section, we present representative outputs from the models evaluated in Section 5. For each example, we include the target word, the accompanying example sentences, and the corresponding explications generated by each model. We then conduct a qualitative analysis of these explications, highlighting in green the one we consider the most effective.

Table 3: Evaluation of NSM explications generated by LLMs for the benchmark set introduced in Section 4. We measure Explication Score (Equation 3, higher is better), Substitutability Score (Equation 2, higher is better), Legality Score (higher is better), Primes Ratio (higher is better), and Molecules Ratio (lower is better). Dictionary Definitions are existing definitions provided from WordNet. Best underlined. While dictionary definitions and Llama-8B, achieve high substitutability, they fail to sufficiently use the NSM primes, lowering their legality and overall scores. DeepNSM models surpass SOTA general LLMs for NSM explication generation despite only having 1B and 8B parameters.

| Model | Explication Score ↑ | Sub. Score ↑ | Legality Score ↑ | Primes Ratio ↑ | Mols. Ratio ↓ |
|---|---|---|---|---|---|
| Dictionary Def. | **13.4** ± .16 | 12.14 ± .08 | -4.7 ± .0082 | 8.0 ± .0008 | 55.1 ± .0010 |
| Llama-3.2-1B-it | **6.1** ± .09 | 6.82 ± .05 | -0.4 ± .01 | 29.2 ± .0008 | 33.2 ± .0008 |
| Llama-3.1-8B-it | **20.0** ± .11 | 7.86 ± .05 | 2.3 ± .01 | 43.9 ± .0008 | 20.9 ± .0006 |
| Gemini-2.0-Flash | **22.2** ± .10 | 6.40 ± .05 | 5.1 ± .01 | 62.1 ± .0006 | 11.5 ± .0004 |
| GPT-4o | **22.9** ± .10 | 6.95 ± .05 | 4.6 ± .01 | 59.9 ± .0006 | 13.8 ± .0005 |
| DeepNSM-1B[†] | **19.7** ± .11 | 5.22 ± .05 | 5.0 ± .01 | 61.1 ± .0009 | 10.9 ± .0005 |
| DeepNSM-8B[†] | **22.2** ± .11 | 6.40 ± .05 | 4.9 ± .01 | 60.0 ± .0008 | 11.5 ± .0004 |
| DeepNSM-1B | **23.2** ± .10 | 7.02 ± .05 | 5.1 ± .01 | 61.9 ± .0007 | 10.6 ± .0004 |
| DeepNSM-8B | **24.6** ± .10 | 7.34 ± .05 | 5.4 ± .01 | 63.9 ± .0007 | 10.4 ± .0004 |

[†]Trained on a dataset with no quality filtering applied.

Table 4: Cross-translatability BLEU scores (↑) on model-generated explications for five low-resource languages from different families: Alur (alz), Kinyarwanda (rw), Dzongkha (dn), Dinka (din), and Abkhaz (ab). Best scores are underlined and bolded.

| Model | BLEU ↑ | | | | |
|---|---|---|---|---|---|
| | alz | rw | dn | din | ab |
| Dictionary Def. | 22.9 ± .13 | 35.1 ± .16 | 19.6 ± .10 | 24.4 ± .13 | 26.9 ± .15 |
| Llama-3.2-1B-it | 29.0 ± .10 | 45.0 ± .12 | 21.0 ± .07 | 25.8 ± .09 | 25.2 ± .08 |
| Llama-3.1-8B-it | 33.2 ± .10 | 50.8 ± .11 | 23.8 ± .08 | 31.8 ± .09 | 27.8 ± .09 |
| Gemini-2.0-Flash | 33.5 ± .07 | 56.3 ± .10 | 23.2 ± .06 | 32.2 ± .07 | 30.1 ± .08 |
| GPT-4o | 31.2 ± .08 | 51.3 ± .11 | 21.6 ± .07 | 31.2 ± .09 | 28.7 ± .08 |
| DeepNSM-1B | 34.5 ± .11 | 56.9 ± .11 | 23.6 ± .07 | 32.6 ± .09 | 31.9 ± .09 |
| DeepNSM-8B | 37.0 ± .10 | 55.6 ± .10 | 23.3 ± .07 | 32.4 ± .08 | 33.2 ± .09 |

Table 5: Cross-translatability Embedding Similarity (↑) on model-generated explications for five low-resource languages from different families: Alur (alz), Kinyarwanda (rw), Dzongkha (dn), Dinka (din), and Abkhaz (ab). Best scores are underlined and bolded.

| Model | Embedding Similarity ↑ | | | | |
|---|---|---|---|---|---|
| | alz | rw | dz | din | ab |
| Dictionary Def. | 69.6 ± .13 | 79.6 ± .10 | 73.3 ± .11 | 69.1 ± .13 | 78.0 ± .10 |
| Llama-3.2-1B-it | 79.3 ± .07 | 88.0 ± .05 | 81.3 ± .06 | 77.3 ± .08 | 84.2 ± .05 |
| Llama-3.1-8B-it | 81.6 ± .07 | 89.8 ± .06 | 81.8 ± .06 | 78.1 ± .07 | 84.0 ± .06 |
| Gemini-2.0-Flash | 84.0 ± .06 | 93.0 ± .03 | 81.7 ± .06 | 78.5 ± .06 | 85.1 ± .05 |
| GPT-4o | 82.0 ± .05 | 91.4 ± .04 | 78.9 ± .07 | 77.4 ± .07 | 84.8 ± .05 |
| DeepNSM-1B | 81.6 ± .07 | 91.4 ± .04 | 80.4 ± .06 | 78.0 ± .07 | 82.7 ± .07 |
| DeepNSM-8B | 84.2 ± .05 | 91.7 ± .04 | 82.0 ± .06 | 79.0 ± .06 | 85.4 ± .05 |

**\<user\>**
Word: **_apologize_**
Examples:
The captain apologized for the malfunctioning engine.
He apologized for losing the team's documents during the crucial meeting.
She apologized for neglecting to RSVP on time for her friend's birthday party.
The customer service representative apologized for the lengthy wait time at the store.
The professor apologized for assigning too many homework assignments in a short span of time.
Paraphrase:

---

**DeepNSM-8B**
someone says something to someone else
because they did something bad to this someone
they say something like this: "I feel something bad because I did this bad thing to you"
they want this someone to know that they feel something bad because of this
they want this someone to not feel something bad because of this

---

**DeepNSM-1B**
I say something to someone
I say something like this:
"I did something
I know that when I did this, you felt something bad
I am saying this because I want you to know that I feel something bad now
I want you to know that I did not want you to feel something bad
I want you to know that I want to do something so that you do not feel something bad"
I say this because I did something that I know can make someone feel something bad

---

**GPT-4o**
someone says words to another person
this someone thinks:
 something bad happened
 this something happened because this someone did something
 this something is bad for the other person
 this someone doesn't want things like this to happen
this other person can feel something good when this other person hears these words

---

**Gemini-2.0-Flash**
say something
say words of one kind
I say something like this because I did something
I think something like this:
"I did something
because of this, something bad happened
I did not want this to happen
I feel something bad now because I did this"
I say these words because I want the person who feels something bad because of this to not feel this bad thing any more

---

**Llama-3.1-8B-Instruct**
say sorry words to someone
because of something that happened, something bad, or not what someone wanted
this is something that someone did or didn't do
someone is responsible for this thing
they feel bad about this thing and think it was not good
they want the person they said sorry to to feel better
this person is thinking: "I made a mistake, and I am sorry"

---

**Llama-3.2-1B-Instruct**
I said sorry
the person that I was in charge of said sorry
he said sorry for losing something important
she said sorry for not saying yes when asked
the person who is in charge said sorry for a long time at this place
the person who has a lot of work said sorry for a long time
the person gave a lot of homework said sorry for a short time
the person in charge said sorry for a lot of work in a short time

---

Figure 4: Model-generated explications for the word "apologize." Both the Llama-3.1-8B-Instruct and Llama-3.2-1B-Instruct models clearly fail to utilize the semantic primes, introducing unnecessary words like "responsible," "homework," "mistake," and containing circularity ("sorry words"). GPT-4o's explication successfully identifies that something bad happened and that the speaker wants the listener to feel better. However, it fails to clearly express the speaker's regret—something like "I feel bad because I did something bad to you"—which is essential for an apology. It also omits that the speaker is saying these words specifically to communicate this feeling of regret (make the other someone "know" they "feel bad"). The explication from Gemini-2.0 Flash is slightly redundant in its opening lines and similarly falls short in expressing the speaker's internal state. Like GPT-4o, it focuses on wanting the listener to stop feeling bad, rather than emphasizing the speaker's desire to communicate their own regret. DeepNSM-1B captures most of the core components of an apology. It reflects a past action that harmed the listener ("I did something ... you felt something bad"), the speaker's awareness of this harm, and their emotional response of regret ("I feel something bad because I did this bad thing to you"). It also notes the speaker's desire to communicate this feeling ("I want you to know that I feel something bad now"). However, it introduces an extra element about doing (not just saying) something to change the listener's feelings, which slightly weakens the clarity of the apology by implying a broader corrective intent. DeepNSM-8B builds on the strengths of 1B while avoiding this ambiguity. It clearly conveys the past harmful action, the speaker's acknowledgment and regret, and the intent to communicate this regret explicitly in order to make the hearer feel better. We believe DeepNSM-8B provides the most accurate and complete explication for this word.

**\<user\>**
Word: **elbow**
Examples:
He injured his elbow while playing basketball.
The anatomist studied the joint structure of the human elbow and its comparison to that of a horse's foreleg.
The dog's elbow gave a sudden pop, as it flexed and extended its forelimb.
The horse's elbow is specially adapted for leverage and strength in running.
Paraphrase:

**Human-Written**
one part of someone's arm
it is a place on one side of the arm
it is hard
the arm has two long parts
one of these two long parts of the arm is on one side of this place
the other long part of the arm is on the other side of this place
because people's arms have this hard part,
one of the two long parts can move when the other long part is not moving

**DeepNSM-8B**
is one part of someone's body
    it is near the middle of someone's arm
    it is where the arm can move in one way
    it is where the arm can bend
    it is not like other parts of the body
    it is a part that makes the arm move in one way

**DeepNSM-1B**
one part of the body
    this part is between the arm and the hand
    it is near the place where the hand touches the body

**GPT-4o**
one part of an arm or leg
this part is between two other parts, they are not the same
these two other parts can move in many ways because of this one part
when this part moves, these two other parts move at the same time

**Gemini-2.0-Flash**
one part of someone's body or one part of another living thing's body
this part is like this:
this part is between two other parts of the body
these two parts are long
this part lets these two long parts move in relation to each other
if this part is not good, someone cannot move these two long parts in a good way

**Llama-3.1-8B-Instruct**
a part of someone/thing's body
this part is a place where one part of this someone/thing connects to another part
this place is not the same as other places in this someone/thing's body
this place is a part of the part that connects two parts of the body: one part goes up and one part goes down
when one part of this someone/thing moves, the other part moves too
because of this, this someone/thing can do many things with this part
when this part of this someone/thing is injured, this someone/thing feels pain because of it

**Llama-3.2-1B-Instruct**
the part of the body above the shoulder
the joint between the leg and the arm
the bone that holds the arm in place
the joint that bends and straightens the arm
when the arm is bent, it touches the ground

Figure 5: Model-generated explications for the word "elbow." Similarly to Figure 4, the Llama models again produce noticeably lower-quality outputs than the rest of the group. While Llama-3.1-8B makes slightly better use of the primes, its description is vague and includes imprecise and potentially misleading information, for example, stating that "one part goes up and one part goes down," which doesn't clearly characterize what an elbow does. GPT-4o's explication has several issues. It introduces the term "leg," likely influenced by the animal examples (e.g., dogs and horses), but an NSM explication should typically define "elbow" specifically in terms of its relation to the human body. The phrase "when this part moves, these two other parts move at the same time" is also vague and fails to clearly capture what the elbow does. Gemini-2.0 Flash's description of the elbow as being "between two long parts" is too vague, and likely is closer to an explication for "knee," making the explication ambiguous and not clearly specific to the concept of "elbow." While DeepNSM-1B provides a much more concise description, locating the elbow at "where the hand touches the body," is not accurate. We identify DeepNSM-8B as providing the clearest and most accurate explication for this word. It introduces simple molecules like "arm," "middle," and "bend" that are comparable in complexity to those used by other models, but it uses them more effectively to clearly convey both the location of the elbow and what it does.

**<user>**
Word: **hypothesis**
Examples:
The chemist proposed a hypothesis suggesting that the catalytic activity of the enzyme could be increased by altering its active site structure. If this hypothesis holds, it could lead to the development of more efficient industrial catalysts.
The medical researcher developed a hypothesis that a new strain of the virus was causing the mysterious outbreak of flu symptoms.
Paraphrase:

**DeepNSM-8B**
someone can think something about something
someone can think like this: "maybe something is like this"
someone can think this because they know some things about something
if someone thinks like this, they can say something about it to other people
if someone thinks like this, they want other people to think about it too
if other people think about it too, they can do something because of it
if they do something, maybe they can know if what someone thought is true

**DeepNSM-1B**
someone thinks about something
they think about it like this:
"I don't know if this is true
maybe this is true
maybe it is not true
I want to know if it is true"
someone can say something about this thing
someone can say something like this:
"I think that this thing can happen because I think this way about it"
someone can say something like this:
"I want to know if this thing is true"

**GPT-4o**
something someone thinks can be like this: "maybe something is like this"
this someone doesn't know if it is true
this someone wants to know if it is true
because of this, this someone does some things to know if it is true or not true

**Gemini-2.0-Flash**
something someone thinks
someone thinks like this: "maybe something is like this"
someone thinks that if something is like this, because of this something else will happen
the someone cannot know if the first thing is true or not
the someone wants to see if the first thing is true
the someone does things to see if the second thing happens
if the second thing happens, the someone will think the first thing is more true
someone says something like this to other people, so other people know what the someone thinks is maybe true

**Llama-3.1-8B-Instruct**
a thought about some things
it is a thought about how some things can be, or about what might happen because of some things
someone says this thought in order to think about it further
if this thought is true, then something else can happen
people can think about this thought and think about what might happen if it is true

**Llama-3.2-1B-Instruct**
a guess, a idea, an assumption, a theory
The chemist proposed a idea that the active part of the enzyme could be changed to make it more effective
the medical researcher came up with a idea that a new kind of virus is causing these symptoms
this idea might work if it's true

Figure 6: Model-generated explications for the word "hypothesis." The word hypothesis refers to a proposition about something unknown that can be tested or disproven through evidence. As in earlier examples, the Llama model outputs can be dismissed. GPT-4o's explication fails to clearly convey falsifiability based on evidence—its phrase "doing some things to know if it is true or not true" lacks the idea that this determination depends on whether certain things happen or not. DeepNSM-8B makes the same omission. DeepNSM-1B's output is weaker still: it does not clearly present the initial proposition (e.g., "this thing is like this") and fails to include any sense of testing or "doing" to evaluate it. We identify Gemini 2.0 Flash as offering the strongest explication for this word. It captures the predictive structure of a hypothesis ("if something is like this, something else will happen"), the testing process ("someone does things to see"), and the dependence on evidence for evaluation ("if this happens, the someone will think the first thing is more true").

**<user>**
Word: **rough**
Examples:
The old oak had a rough bark.
rough skin
Paraphrase:

**GPT-4o**
when you touch this thing, you feel something bad
there are many small points and hard places on the outside of this thing
it is not like touching a smooth, good thing
people can feel it like this when they make contact with it

**DeepNSM-8B**
something is like this:
when someone touches this thing, this someone can feel something bad
this someone can feel like this: "this is not good"
this is because there are many small things on the outside of this thing
these small things are not the same
because of this, when someone touches this thing, the small things can touch someone's body at the same time

**Gemini-2.0-Flash**
when someone touches it with one's body, it does not feel good to someone
when someone touches it with one's body, someone can feel many little things
these little things are not all the same
they are not like this: all of them are not above the part that someone touches
they are not like this: all of them are not below the part that someone touches
because of this, it does not feel good to someone when someone touches it with one's body

**DeepNSM-1B**
if something is like this, it is not like this: if someone touches it, this someone can feel many small things that are not good to touch
because of this, if someone touches it, this someone can feel something bad

**Llama-3.1-8B-Instruct**
not smooth
the surface of this thing is not smooth
the surface of this part of this person's body is not smooth

**Llama-3.2-1B-Instruct**
hard to touch
rough to touch
rough on my skin

Figure 7: Model-generated explications for the word "rough." As with previous examples, the Llama-generated explications can be clearly set aside. GPT-4o's version introduces non-prime terms like "contact" and "points," which add unnecessary complexity, but it still successfully conveys the unpleasant feeling caused by touching a surface with many small, hard projections. Gemini-2.0 Flash attempts to express a similar idea, but its phrasing—such as referencing things "not above/below the part"—introduces confusion, as these spatial qualifiers don't clearly relate to the sensation of roughness. DeepNSM-1B includes a critical error by stating that things like this do not have small things on the outside, which is the opposite of what "rough" means. This contradiction makes the explication ineffective. In contrast, DeepNSM-8B avoids these issues: it captures the same descriptive meaning as GPT-4o without introducing non-prime terms or misleading spatial language. We identify DeepNSM-8B's explication as the most accurate and effective for this word.

# E  PROMPTS

**\<user\>**
Word: **emergency**
Examples:
It was an **emergency** situation that demanded immediate evacuation
The traffic accident was a major **emergency**
Paraphrase:

**\<assistant\>**
something very bad is happening to people in this place
because of this, someone there has to do something now
they have to do something not like at other times
they have to do it very quickly
if they don't do this, something very, very bad will happen to people in this place
it will be very bad for people in this place

Figure 8: Envisioned User-Assistant Interaction

**\<system\>**
Generate 4 distinct example sentences using the specified word strictly according to its defined sense. Ensure each sentence appears on a new line, without using numerical or bullet point separators. Only the sentences should be provided as output.

**\<user\>**
Word: pull, used in the sense 'cause to move by pulling' such as 'pull a sled'

**\<assistant\>**
She helped pull the heavy boxes out of the storage room.
The child enjoyed pulling her friends on the sled down the snowy hill.
The team worked together to pull the stuck car out of the muddy ditch.
The athlete had to pull herself up onto the gymnastics beam with a burst of strength.

Figure 9: Prompt for generating additional usage examples.

**<system>**
Generate 4 distinct example paragraphs using the specified word according to its defined sense, ensuring the word is masked as <UNK>.
- The word must only appear once in the paragraph.
- Each paragraph should be 3 sentences long.
- Ensure there is no use of numbers or bullets to separate paragraphs. Each should appear on a new line.
- Ensure the surrounding paragraph is highly ambiguous and could not be used to guess the masked word.
Only provide the paragraphs as the output.

**<user>**
Word: pull, used in the sense 'cause to move by pulling' such as 'pull a sled'

**<assistant>**
The old barn creaked as the wind howled through the broken shutters, hinting at a forgotten past. As I glanced at the nearby fields, my imagination wandered to the weighty tasks that once occupied the land. There was a moment when I felt compelled to <UNK> something heavy across the uneven terrain.
… (x4)

Figure 10: Prompt for generating additional ambiguous examples for the substitutability testing described in Section 3.2.

**<system>**
You are a linguist specializing in semantic analysis using the Natural Semantic Metalanguage (NSM) approach. NSM is a linguistic theory that reduces lexicons down to a set of universal semantic primes. You will be given a word and example passages where the word is used. Your task is to paraphrase the word's meaning using the NSM primes. Here are guidelines for your paraphrase:
- Use NSM primes and simple words from ordinary language.
- The paraphrase should be exhaustive and should portray the full meaning of the word being analyzed.
- The paraphrase must be able to replace the original word in ALL of the examples without changing the meaning.
- Do not use the original word or a close synonym to it.
- Avoid obscurity, circularity, and do not introduce words that are more complex than the original.
Do not use logical symbols or abbreviations.

List of NSM Primes:
I, you, someone, people, something/thing, body, kind, part, the same, other/else/another, one, two, some, all, much/many, little/few, good, bad, big, small, think, know, want, don't want, feel, see, hear, say, words, true, do, happen, move, be (somewhere), there is, be (someone/something), (is) mine, live, die, when/time, now, before, after, a long time, a short time, for some time, moment, where/place, here, above, below, far, near, side, inside, touch (contact), not, maybe, can, because, if, very, more, like/as/way.

Figure 11: Explication Generation System Prompt.

**<user>**
Word: land
Examples:
the promised land
a land flowing with milk and honey
This land is your land, this land is my land
Paraphrase:

**<assistant>**
a big place
it is one part of the earth
it is not like other parts of the earth
it is like this in this place:
      places of many kinds are parts of this place
      things of many kinds grow in the ground there
      living creatures of many kinds live there
because of this, many people can live in this place
people can think about them like this: "these people are people of one kind"

**<user>**
Word: we
Examples:
We saw a great movie last night.
We were here first.
We like to go to the ice cream store.
Paraphrase:

**<assistant>**
all these someones, I am one of all these someones
when I say this, I'm thinking about all these someones in the same way"

**<user>**
Word: virus
Examples:
I must've picked up a virus.
Colds are caused by a virus.
It's only a matter of time before such a virus adapts itself to spread more easily from person to person and cause a severe worldwide outbreak.
Paraphrase:

**<assistant>**
something of one kind
there are many kinds of things of this kind
when there is something of this kind somewhere, there are many very very small things of this kind there, people can't see them
there can be many such things in people's bodies at some times
when there are many of these very very small things in someone's body, a very short time after there can be many of them in the bodies of other people in the same place
when they are in someone's body, something bad can happen in this someone's body because of it
people can think about things of this kind like this: "these things do very bad things to people's bodies"

Figure 12: Few-Shot Examples used for the Explication Generation Prompt.

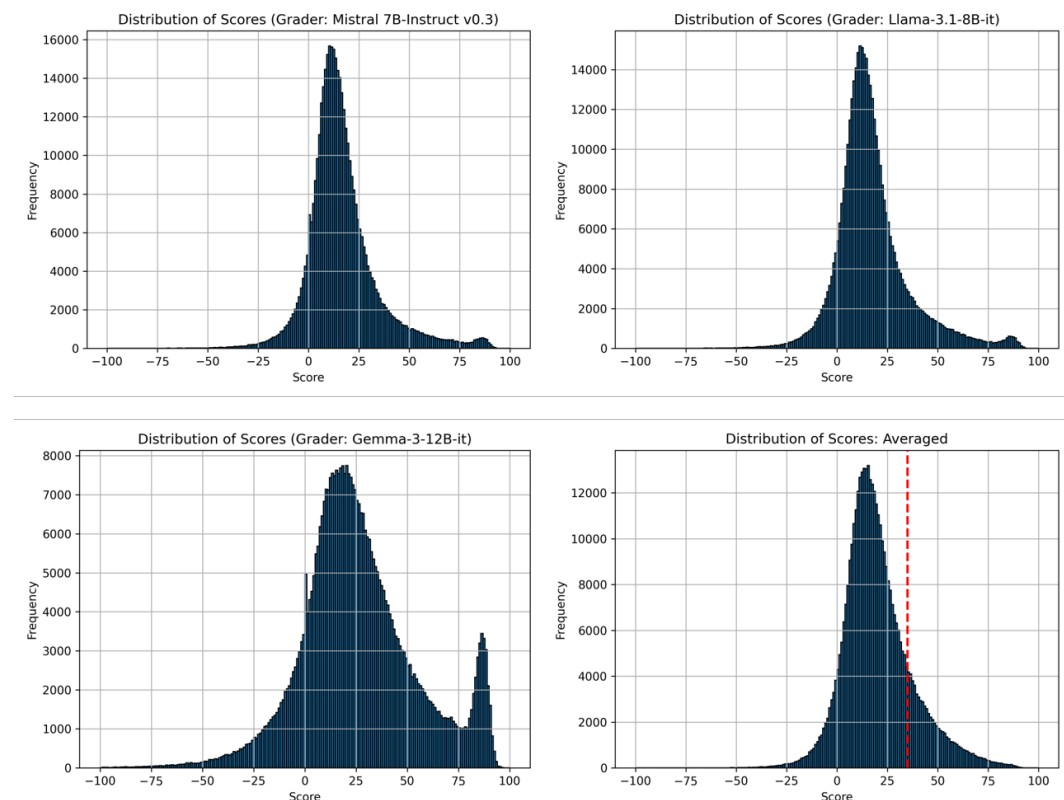

Figure 13: Distribution of Explication Scores assigned by different "grader" LLMs during substitutability testing, evaluating all candidate explications generated through the dataset creation process described in Section 4. The bottom right plot shows the distribution of scores after averaging the scores given by all graders. The red line indicates the cutoff threshold of $\geq 35$ from Section 4.

Table 6: Pairwise Disagreement between grader models from the dataset construction process described in Section 4. We evaluate how much the three grader models used in the main text—Llama-3.1-8B, Gemma-3-12B, and Mistral-7B-v0.3—disagree with one another by measuring the differences in the substitutability scores each model assigns. Agreement expected by random chance is shown in parentheses.

| | Pairwise Disagreement % | | |
| Model Pair | <5.0 | <10.0 | < 15.0 |
| --- | --- | --- | --- |
| Mistral + Llama | 57.9% (30.8%) | 79.4% (57.1%) | 89.4% (76.4%) |
| Llama + Gemma | 35.4% (22.3%) | 60.6% (43.0%) | 76.5% (60.7%) |
| Gemma + Mistral | 34.8% (22.5%) | 60.0% (43.2%) | 75.9% (60.8%) |

# F  ROBUSTNESS OF GRADER LLMS

Figure 13 in the appendix shows the distribution of scores assigned by each grader model. To further evaluate robustness across grader models, we analyzed how often the models disagreed on the same explication, seen in Table 6. We found that in 84% of cases, the score difference between models was less than 10 points (most scores fall between -10 and 40). Additionally, we also assessed pairwise agreement using binary thresholds of the substitutability score. As shown in the table below, Mistral and Llama were the most aligned, while Gemma tended to give higher scores and showed slightly lower agreement with the others. Still, all model pairs showed much higher agreement than expected by chance, defined as sampling scores randomly from each model's distribution (shown in

Table 7: List of open lexical resources that can be used to extend this work towards other languages.

| Name | Word Count (entries) | Language |
|---|---|---|
| WordNet | 120,600 synsets | English |
| Wiktionary | 853,000 lemma entries | Multilingual |
| BabelNet | 23,000,000 synsets | Multilingual (600 langs.) |
| Spanish WordNet | 37,876 synsets | Spanish |
| Hindi Wordnet | 26,208 synsets | Hindi |
| plWordNet | 347,000 synsets | Polish |
| WOLF | 59,000 synsets | French |

parentheses). These findings indicate that, while scoring tendencies vary slightly across models, the substitutability metric remains relatively robust and consistent.

## G    EXTENDING THIS WORK TO OTHER LANGUAGES

While this work focuses on paraphrasing English texts into the semantic primes, the methods we propose are readily extendable to other high-resource languages. Even in cases where parallel corpora are limited, the approach remains viable as long as the underlying language models have been trained on sufficient text in the target language. This opens up immediate opportunities for researchers and practitioners aiming to translate text in a high-resource language into a low-resource one. Specifically, high-resource paraphrases into semantic primes can serve as an intermediate representation that is more easily translated into low-resource languages, potentially reducing ambiguity and improving semantic fidelity.

Additionally, our work opens valuable opportunities for paraphrasing texts written in low-resource languages (LRLs) into the semantic primes—a task that is very challenging due to limited data and linguistic documentation. One promising direction is to leverage English-to-primes models like DeepNSM to first convert large English corpora into semantic primes. These prime-level representations can then be used to guide models training on LRLs, especially once the primes have been identified and validated in the target language. For example, models could learn to paraphrase unknown words in the LRL by analyzing how they appear near known prime expressions—effectively applying an informed form of distributional semantics grounded in semantic primes. However, this approach risks introducing biases from English-specific prime usage patterns, highlighting the need for further cross-linguistic study.

Alternatively, the evaluation methods proposed in this work can serve as optimization signals or auxiliary objectives for training or fine-tuning models directly on LRL text. In cases where training data is sparse, fluent speakers of the target language could also use the paraphrased dataset as a foundation, combining our automatic techniques with human annotation to refine and validate prime-level paraphrases, leading to high-quality NSM datasets in a wide range of languages.

Additionally, this work is not solely dependent on WordNet. Table 7 shows a (non-exhaustive) list of many similar resources which the approach demonstrated in this paper could be applied to. In addition to hand-built ontologies/structured data, our method, with slight modifications, can be bootstrapped directly from unstructured text corpora rather than relying on structured lexical resources like WordNet as a seed.

## H    APPLYING NSM FOR DOWNSTREAM TASKS

We provide some examples of how using NSM can aid in two example downstream tasks: cross-lingual document retrieval on Swahili from the CIRAL dataset (4), and question-answering (QA) on the Belebele dataset (8).

### H.1 Document Retrieval

For document retrieval, we compared two prompts to GPT-4.1: a baseline prompt asked the model to pick the most relevant document and explain its reasoning. An NSM-based prompt first decomposed the user query into semantic primes and used this paraphrase to guide the retrieval reasoning.

#### H.1.1 Example 1

**Query:** *What are some football leagues does Manchester United play in?*

**Document A (Relevant):** *"...Kipute cha Klabu Bingwa Ulaya (UEFA)...", "...mechi ya Carabao Cup..."*

**Document B (Non-Relevant):** *"Kwa sasa Manchester United wanashika nafasi ya 12 kwenye msimamo wa Ligi baada ya kushinda..."*

**Baseline Prompt Result:** The baseline prompt directed the model to rank the documents and explain. With this prompt, GPT-4.1 incorrectly chose Document B, stating:

*"Document B [...] directly indicates that Manchester United is participating in a league. While it doesn't name the league explicitly, it's more directly related to the query..." Whereas Document A was dismissed as "primarily discuss[ing] friendly matches [...] and does not explicitly state what leagues Manchester United plays in."*

**NSM-Guided Prompt Result:** The NSM-enhanced prompt first broke down the query:

*"football leagues" = kinds of groups/places where people play these games (football), not one game, but many games... "Manchester United play in" = people of Manchester United do this: they are inside such a group and do football with other people, not one time..."*

It then analyzed the documents accordingly. For Document A, the model recognized:

*"...mentions UEFA Champions League, Carabao Cup, and friendly matches... [this gives] multiple examples of 'kinds of things (groups/leagues)' where Manchester United plays..."*

For Document B, the analysis concluded:

*"...only gives current standing in 'Ligi' (singular) without specifying the kind or name of league..."*

Final decision:

*"Document A is more relevant because it gives more examples of 'kinds of places' where Manchester United plays football."*

#### H.1.2 Example 2

**Query:** *What is the national airline of Ethiopia?*

**Document A (Relevant):** *"...shirika la ndege la Ethiopia..."* (mentioned as the largest in Africa, in comparison to South African Airways)

**Document B (Non-Relevant, text omitted for brevity):** Mentions "Ethiopian Airline" multiple times in the context of an airplane crash, without clarifying its status as a national carrier.

**Baseline Prompt Result:** With the baseline prompt, GPT-4.1 incorrectly selected Document B, due to repeated mentions of the phrase "Ethiopian Airline" in connection with a plane crash. However, no statement in Document B connects Ethiopian Airlines to the country Ethiopia as its primary or national carrier.

**NSM-Guided Prompt Result:** The NSM-enhanced prompt first broke down the query as follows:

*"I want to know: there is one big company, this company does something with planes, this company is from a place people call Ethiopia, this company is a company of all people there. What is the name of this company?"*

It then evaluated the documents based on this structure. For Document A, the model recognized:

*"...mentions 'shirika la ndege la Ethiopia' — the airline of Ethiopia — and compares it to South African Airways, implying it is the national carrier..."*

For Document B, the analysis concluded:

*"...only gives current standing in 'Ligi' (singular) without specifying the kind or name of league..."*

Final decision:

*"Document A is more relevant because it refers to the airline using the phrase 'shirika la ndege la Ethiopia', directly aligning with the idea of a company that is from Ethiopia and belongs to all people there."*

### H.2  QUESTION-ANSWERING

Similarly, augmenting LLMs with NSM explications can also help models to reason better for QA tasks.

#### H.2.1  EXAMPLE 1

**Question:** *What did Hershey and Chase use to implant their DNA?*

**Passage (Jingpo):** *" ...Hershey hte Chase gaw phage, dai nrai viruses, ni hpe shanhte DNA bacterium kata de bang ai hta lang ma ai..."* (...Hershey and Chase used phages, a kind of virus, to insert DNA into bacteria. They used radioactive phosphorus to label DNA and radioactive sulfur to label the protein coat...)

**Answer Choices:**

1 Radioactive phosphorus

2 Genetic rai

3 Radioactive sulfur

4 Viruses (Correct Answer)

**Baseline Prompt Result:** With the standard prompt, GPT-4.1 chose Answer 1 (Radioactive phosphorus) and gave the following justification:

*"The question asks what Hershey and Chase used to implant their DNA. The passage says they used radioactive phosphorus in the DNA and radioactive sulfur in the protein coat. This allowed them to track which molecule entered the bacteria. Since radioactive phosphorus was used to label the DNA, it is the correct answer."*

Here, the model equated "implant" with "label" or "trace," rather than "putting DNA inside" another entity. While it correctly identified phosphorus as a DNA label, it misunderstood the functional role of viruses in the process.

**NSM-Guided Prompt Result:** The NSM-enhanced prompt began with a breakdown of key terms in the question:

1. *"implant"* → *to put something inside something else, so that it is there*
2. *"DNA"* → *something inside living things (people, animals, plants) that makes them what they are*
3. *"use"* → *to do something with something else, to make something happen*

The question was then rephrased internally as:

*"What thing did Hershey and Chase do something with, so that their DNA was inside another thing?"*

Using this lens, GPT-4.1 re-evaluated the passage and provided this reasoning:

*"The passage says: 'Hershey and Chase used phage, some kinds of viruses, to make DNA go into a bacterium.' This directly answers the question if we understand 'implant' to mean 'put something inside something else.' The radioactive phosphorus was used to see where the DNA went—it did not*

*put the DNA into the bacteria. The phage (virus) is what did that. The sulfur was only used to mark the protein part. So the only thing that matches the idea of doing something to make the DNA go inside is the virus."*

The model now correctly identified the role of phages as the active agents of DNA delivery—something the baseline model failed to do.

Final Decision:

*"Viruses are the thing Hershey and Chase used to put their DNA inside something else. Phosphorus only showed where the DNA was, but viruses are what did the putting."*

