# OpenReview forum: "Towards Universal Semantics with Large Language Models"
_ICLR.cc/2026/Conference — Submitted to ICLR 2026_

### Official Review · Reviewer_dV16 · 2025-10-30

**Soundness:** 3
**Presentation:** 3
**Contribution:** 3
**Rating:** 6
**Confidence:** 4

**Summary:**

This paper focuses on a study that uses Large Language Models (LLMs) to address the issue of "linguistic semantic generalization". Simply put, it aims to enable more accurate and consistent expression and translation of meanings across different languages. The core approach is to integrate a theory called "Natural Semantic Metalanguage (NSM)".

The NSM theory posits that all languages share a set of "universal semantic primitives" (such as "I", "you", "good", "bad", "time"—these are the simplest words). No matter how complex a word is, it can be rephrased using these "primitives". For example, the word "ill" can be explained as "something bad is happening inside a person’s body, and the person can feel this badness". This type of explanation is called an "explication". It allows for accurate cross-linguistic translation without unnecessary complexity.

For instance, when translating the English word "color" into a language that has no dedicated color terms (e.g., Warlpiri), the approach first uses NSM to "explicate" "color" into simple sentences. This makes the subsequent translation much easier.

**Strengths:**

It is the first study to combine Large Language Models (LLMs) with the Natural Semantic Metalanguage (NSM) framework. This breaks the limitation of traditional NSM explications, which relied on manual generation. No previous research has explored the application of LLMs in generating NSM explications. This study fills this gap and provides a new technical pathway for universal semantic representation.

It proposes an automated evaluation method for NSM explications, covering three core dimensions: validity, descriptive accuracy, and cross-linguistic translatability. Additionally, it constructs the first custom dataset for this task (containing approximately 44,000 entries of "word-example-explanation") and develops the DeepNSM fine-tuning model.

The DeepNSM model outperforms general-purpose LLMs such as GPT-4o and Gemini in key metrics. In cross-linguistic translatability tests for low-resource languages (e.g., Aruul, Abkhaz), it also achieves better performance in indicators like BLEU scores. This provides a feasible solution for practical needs such as low-resource language translation and cross-cultural communication.

**Weaknesses:**

The example sentences and candidate explications in the dataset mainly rely on LLM generation. Although automated screening was conducted, the linguistic biases of LLMs (e.g., English-centrism) may be transferred to the dataset. This affects the accuracy of the model’s explications for words from non-English cultural backgrounds. Furthermore, manual verification only covers 149 entries in the test set, with no large-scale manual validation of the training set.

The study primarily builds the dataset and trains the model based on English vocabulary. While it mentions that the approach can be extended to other languages, it does not provide experimental validation for non-English languages (e.g., it does not test the effectiveness of NSM explication generation for languages like Chinese or Spanish). This makes it impossible to confirm the method’s generalization ability in multilingual scenarios. Additionally, the testing of low-resource languages is limited to 5 languages, with a small sample size.

The dynamic adaptability of semantic primitives is not discussed. The semantic primitives of the NSM framework are regarded as "universal and fixed", but the paper does not explore whether these primitives need to be adjusted according to linguistic or cultural differences. For example, some low-resource languages may have unique core semantic units. It also fails to analyze the adaptability of semantic primitives when the model processes culture-specific words (e.g., emotion-related words or words specific to a particular region).

**Questions:**

1. Since the example sentences and candidate explications in the dataset rely on LLM generation, how can the impact of LLMs’ linguistic or cultural biases on dataset quality be quantified and mitigated?

2. For non-English languages (e.g., Chinese, Swahili), is it necessary to adjust the selection of semantic primitives when building the dataset? What is the specific logic behind such adjustments?

3. The paper mentions that the method can be extended to other languages, but it does not provide experimental data for non-English languages. Can additional experiments on NSM explication generation for languages like Chinese or Spanish be supplemented to verify the multilingual generalization ability? For low-resource languages without clear validation of semantic primitives (e.g., some African tribal languages), how can the equivalence of semantic primitives be determined?

4. The paper does not compare the effectiveness of combining LLMs with other semantic frameworks (e.g., FrameNet, WordNet semantic decomposition) versus combining LLMs with NSM. Why was NSM chosen over other frameworks, and how were its core advantages quantified in the experiments?

---

> ### Author Response · Authors · 2025-12-02
>
> ## "Since the example sentences and candidate explications in the dataset rely on LLM generation…”
>
> To explore how LLM bias surfaces, we prompt GPT-4o (English-centric) to paraphrase “home” and DeepSeek (Chinese-trained) to paraphrase “家” (translated into English as “home”)
>
> ### GPT-4o
>
> A place.
> Someone lives in this place.
> This someone can say:
> “I feel something good when I am in this place.
> I don’t want to be in other places for a long time.
> I want to be in this place for a long time.
> I can do many things in this place.
> Things of mine are in this place.
> I can be the way I want to be in this place.
> Because of this, this place is not like other places.”
> People can think about this place and feel something good.
> People can say: “This is my place.”
>
> ### DeepSeek-V3
>
> 这是一个地方，有人（们）一直住在这里，这些人是一起的（像同一类）。这个地方是他们的，他们想要在这里，在这里他们觉得好。很多时候，这些人中有一个是小的（孩子），别的是大的（父母）。这些人在这里做很多事，在这里生活。他们说"这是我们的地方"。当不在这个地方时，他们想要回到这里。
>
> _Translation: This is a place where someone(s) always lives, these people are together (like the same kind). This place is theirs, they want to be here, they feel good here. Often one of these people is small (children) and the other is big (parents). These people do a lot of things here, live here. They say "this is our place". When not in this place, they want to come back here._
>
> GPT-4o biases towards personal autonomy and ownership, describing home through individual experience, freedom, and possessions. DeepSeek instead centers family and shared living, presenting home as a collective space defined by relational ties. We see similar divergences for other culturally loaded concepts such as “face” and “面子.”
>
> Quantifying or mitigating these biases is nontrivial, but NSM may make it more tractable than in typical semantic generation tasks. One possible approach is to construct a small set of culturally sensitive target words and prompt different models to produce “per-language” NSM paraphrases. Because NSM uses a fixed inventory of about sixty primes, the resulting explications can be compared directly against expert-written explications for each language variant. Simple lexical- or structural-similarity measures over the primes can then show which linguistic or cultural background a model’s paraphrases most closely align with. This would provide a practical way to estimate both the magnitude and the direction of model bias and offer a basis for potential mitigation strategies, such as balancing the training data to reduce overrepresentation.
>
> ## “For non-English languages (e.g., Chinese, Swahili), is it necessary…”
>
> A core premise of the NSM framework is that the set of semantic primitives must remain invariant across languages, so they are not adjusted on a per-language basis. What varies is only how each prime is lexicalized in a given language. Introducing different sets of primitives for different languages would undermine one of the central strengths of NSM—its ability to support cross-linguistic semantic comparison using an invariant set of shared units.
>
> ## The paper mentions that the method can be extended to other languages, but it does not provide experimental data for non-English languages…”
>
> We agree that extending generation to other languages is an important direction. The method is naturally extendable because the lexical resources we rely on (e.g., WordNet) have counterparts in other languages, as noted in Section G of the Appendix. In this paper, however, our focus is on establishing that high-quality NSM explications can be generated at all and on presenting a concrete approach for doing so. Once this foundation is in place, the multilingual extensions outlined in Table 7 and Section G can be pursued in subsequent work. As for validating the equivalence of NSM primes in low-resource languages, this typically requires linguistic fieldwork (that is, working directly with native speakers). The following source has many examples for how NSM is applied in such fieldwork settings (see starting at Section 3.2):
> - Goddard, Cliff, and Anna Wierzbicka. "Semantic fieldwork and lexical universals." Studies in Language. International Journal sponsored by the Foundation “Foundations of Language” 38.1 (2014): 80-127.
>
> ## "The paper does not compare the effectiveness of combining LLMs with other semantic frameworks…”
>
> We agree that this is important to compare, however, each of those frameworks is different and would require its own dedicated pipeline, dataset construction, representation scheme, and evaluation setup. Developing and validating each of those systems would be a substantial project in its own right. Our focus here is to investigate whether NSM-style explications can be generated with high quality at all, and to build the modeling and data infrastructure needed to do so. Once that foundation exists, comparisons with other frameworks become feasible, but each framework would most likely require its own unique pipeline rather than a simple drop-in substitution with DeepNSM’s.

---

### Official Review · Reviewer_JpBr · 2025-10-30

**Soundness:** 3
**Presentation:** 3
**Contribution:** 3
**Rating:** 8
**Confidence:** 4

**Summary:**

This paper presents the first study on using large language models to automate the generation of "explications" within the Natural Semantic Metalanguage (NSM) framework. NSM is a linguistic theory proposing that any word's meaning can be paraphrased using a small, universal set of "semantic primes," offering a path to universally translatable semantic representations. The authors identify the slow, manual creation of these explications as the primary bottleneck to NSM's adoption in NLP. To address this, the paper introduces three core contributions: (1) a novel set of automatic evaluation metrics to assess explication quality, including "legality" (prime usage) and "descriptive accuracy" (a clever "substitutability test" using LLM log-probabilities); (2) a new, high-quality dataset of ~44,000 explications, bootstrapped by generating candidates with an LLM and filtering them using their proposed metrics; and (3) fine-tuned 1B and 8B models ("DeepNSM") that outperform strong baselines, including GPT-4o, on this new task. The authors demonstrate that the resulting NSM explications show significantly higher cross-translatability into low-resource languages than standard dictionary definitions.

**Strengths:**

This is an excellent and thoughtfully constructed paper.

* Originality: The paper's originality is outstanding. It is, to my knowledge, the very first work to formally bridge the gap between modern LLMs and the well-established (in linguistics) NSM framework. This is a novel and exciting problem formulation. The authors don't just apply an LLM to a task; they propose a complete "stack" for a new sub-field: evaluation, data, and models.

* Quality: The methodological quality is very high. The authors correctly identified that the primary challenges were the lack of evaluation metrics and data, not just the lack of a model.

* The proposed "Substitutability Score" (Section 3.2) is the standout contribution. It is a highly creative and intelligent solution for automatically evaluating a complex semantic property. Using an LLM as a "grader" and decomposing the score into baseline accuracy (Δ_baseline), minimality (Δ_min), and entailment capture (Δ_ent) is a very robust approach.

* The "generate-and-filter" pipeline for dataset creation (Section 4) is also excellent. Using their own newly-defined metrics to filter the dataset is a very smart way to bootstrap quality, and the ablation study (comparing DeepNSM to its unfiltered-data-trained variant in Table 1) clearly demonstrates the value of this filtering step.

* Clarity: The paper is exceptionally well-written. It takes a concept (NSM) that is likely foreign to most of the ICLR community and explains it clearly, concisely, and with effective examples (Figure 1). The entire paper is logically structured, and the figures (especially Figure 3, illustrating the data pipeline) are highly effective.

* Significance: The potential impact of this work is high. A scalable, grounded, and interpretable universal semantic representation could have major implications for low-resource NLP, machine translation, cross-lingual reasoning, and AI safety/interpretability. By open-sourcing the metrics, dataset, and models, the authors have provided all the necessary tools for the community to build upon this work. The strong cross-translatability results (Table 2) are a powerful proof of concept for the "universal semantics" claim.

**Weaknesses:**

* **Validation of Proposed Metrics**: The entire paper's success hinges on the validity of the new automatic evaluation metrics (Legality and Substitutability). The authors briefly state that "Metrics Align with Qualitative Judgements" (Section 5), but this is presented as a summary (e.g., "DeepNSM models received top rankings 46% of the time"). This is insufficient. A formal, quantitative correlation analysis (e.g., Pearson or Spearman) between the "Explication Score" and blind human ratings is necessary to truly validate that this metric is a reliable proxy for human-judged quality.

* **Sensitivity of the Substitutability Score**: The Substitutability Score is a complex formula combining three deltas and averaged across multiple "grader LLMs." The paper does not provide an ablation or sensitivity analysis on this. For instance, how sensitive is the score to the choice of grader LLMs? What is the contribution of each component (Δ_baseline, Δ_min, Δ_ent) to the final score?

**Questions:**

* Could the authors please provide a quantitative correlation (e.GET., Pearson's r) between the proposed "Explication Score" and human-annotated quality scores? This would significantly strengthen the trust in the paper's central claim, which relies on this new metric.

* Would it be possible to provide a breakdown of the Substitutability Score into its components (Δ_baseline, Δ_min, Δ_ent) for the models in Table 1? This would provide crucial insight into the nature of the DeepNSM models' improvement.

---

> ### Author Response · Authors · 2025-12-02
>
> We appreciate the reviewer taking their time to read and review the paper, and are happy to provide the additional analysis they have requested.
>
> ## “A formal, quantitative correlation analysis (e.g., Pearson or Spearman)... is necessary…”
>
> We use Spearman correlation to compare the human rankings with the rankings produced by the Explication scores. Concretely, we sort the model-generated explications by their Explication score and then compare this ordering to the blinded human rankings. We observed a Spearman correlation of 0.416.
> For reference, Spearman values close to 1 typically occur on tasks with a clear, objective ordering and very high human agreement (e.g., ranking items by numerical value or explicit length). However, for many linguistic evaluation tasks where there are multiple defensible “correct” answers, such as semantic similarity rating, text quality judgments, or summarization evaluation, Spearman correlations between judgments are typically well below 1 and often only moderate, reflecting human variation on the task. For example, [A] shows human annotator agreement at 0.67 for word-similarity judgements, while [B] shows inter-annotator agreement on certain semantic clustering tasks as low as 0.19.
> Given this context, our observed score of 0.416 shows a real, positive correlation between the Explication Score metric and human judgements, and falls within the range typically seen on tasks of this type while still being well above random chance (0). That said, more work is needed to measure human-to-human agreement specifically for NSM paraphrase evaluation so we can determine how much headroom for improvement actually exists. Lastly, we have revised Section 5 to include this correlation analysis.
>
> ## “The Substitutability Score is a complex formula…”
>
> Figure 13 in the appendix shows the distribution of scores assigned by each grader model. To evaluate robustness across grader models, we analyzed how often the models disagreed on the same explication. The results can be seen in Table 6. We found that in 84% of cases, the score difference between models was less than 10 points (most scores fall between -10 and 40). Additionally, we also assessed pairwise agreement using binary thresholds of the substitutability score. As shown in the table below, Mistral and Llama were the most aligned, while Gemma tended to give higher scores and showed slightly lower agreement with the others. Still, all model pairs showed much higher agreement than expected by chance, defined as sampling scores randomly from each model’s distribution (shown in parentheses). These findings indicate that, while scoring tendencies vary slightly across models, the substitutability metric remains relatively robust and consistent. We have revised the manuscript to include this extra analysis in Section F of the Appendix.
>
> # References
>
> A: https://fh295.github.io/simlex.html
>
> B: https://direct.mit.edu/coli/article/47/1/69/97331/Semantic-Data-Set-Construction-from-Human

---

### Official Review · Reviewer_Mafg · 2025-11-03

**Soundness:** 1
**Presentation:** 2
**Contribution:** 1
**Rating:** 2
**Confidence:** 3

**Summary:**

This paper argues that the framework of Natural Semantic Metalanguage (NSM) and the explications of word meanings based on NSM semantic primitives would be useful in NLP applications. They explore whether LLMs can be applied to generate good NSM explications, since usually NSM explication generation by human experts is labor intensive. The question of whether NSM explications are actually useful for NLP applications is a separate one, and the paper also touches on this via a low-resource translation experiment. I think my major issue with this paper is conceptual, having to do with buying into the premise that NSM is a useful framework, although I have several methodological concerns as well, one of the major concerns being the lack of human verification of whether the explications actually do capture the original lexical meaning.

**Strengths:**

The motivation behind the work - more generalizable methods to low resource languages - is something that the field should care about more.

**Weaknesses:**

My main issue with this work is that I have a difficulty convincing myself that the theoretical grounding that the work is built upon is sound and the approach of NSM primitive-based explication is actually a useful way to approach things. I confer that this may be partly due to my own background and priors, and would be open to discussions. Here are some thoughts in this regard:
- Overall framing issues and lack of citations to work being criticized: I don't have a guess about what background of the authors are, but as a self-identified linguist, a lot of the claims and statements in this paper don't sit well with me. For instance, from the very first sentence: "Semantics, the study of word meaning" --- I cannot say I agree with reducing "semantics" to the study of word meaning; calling dictionary definitions a "conventional semantic approach" also seems puzzling to me. These claims are also made without references, so it's hard for the readers to evaluate where such claims are even coming from. Furthermore, even the NSM literature isn't being properly engaged with: e.g., Figure 1a) seems to be just taken from Wikipedia without citing it; the Wikipedia article cites a book as a source for it, which isn't cited in the paper.
- The validity of NSM as a theory of meaning: I also was not sure I was convinced about the status of Natural Semantic Metalanguage as a valid theory or NSM primitives as a valid theoretical construct. From the perspective of someone who's not directly aware of the NSM theoretical enterprise but have linguistics training, I'm not sure the fundamental assumption of NSM as described in the paper, "These primes are considered primitive because they represent fundamental semantic units that cannot be defined in simpler terms" is entirely correct based on the list of natural semantic primes proposed. For instance, concepts like live/die could be expressed in terms of one another by negating one ("NOT" is a primitive, after all). The table lists "WANT" and "DON'T WANT" as primitives, and given my previous comment I think one can easily see why it's problematic for the claim that "semantic primes are fundamental semantic units that cannot be defined in simpler terms".
- I also wasn't sure if the NSM explication in in Figure 1b is (1) correct and (2) actually useful. One general issue I saw (from this example and Figure 9, Appendix E) is what NSM explications in this paper are explications of. Are they explications of the standalone lexical meaning, or of the full expression that contains the lexical item? For example, in the paraphrase example given in Figure 1b supposed to be corresponding just to the meaning of the word "sick"? This definitely cannot be an paraphrase of the standalone lexical meaning, because it contains information about the gender of the person who is feeling sick by using the word "she". In general, it seemed to me that NSM explications (based on the examples given in the paper) are neither paraphrases of the lexical meaning or the full expression containing them. If the former, the current explications are incorrect because they add meanings that are not present in the lexical meaning. If the latter, the current explications miss a lot of information present in the expressions (e.g., in the "emergency" example in Figure 9, the paraphrase misses information about the traffic accident).
- These observations makes me doubt the theoretical groundings and empirical validity of the framework itself. However, independently of these concerns, it is not impossible that the proposed approach of paraphrases using simpler concepts is useful for certain application scenarios. But if I were to be convinced on that level of utility, I would like to see a paraphrase of the meanings of words/phrases like "neural networks" or something simpler like "cats" or "green" via the primes in Figure 1a, since the claims seems to be that this IS an exhaustive list of semantic primes in English and I personally have a hard time seeing these expressed in those terms. Color is even mentioned as a domain that would benefit from NSM in Section 2.2, but I think examples would really help.
- There is also a question of whether the NSM explications will actually be useful to humans, as this is stated as one of the goals (L150: "Models that generate text using semantic primes could make the outputs of LLMs and other AI models accessible to speakers of all language"). As a speaker of English, I'm really not sure whether the offered NSM explication in Figure 1b is useful or even comprehensible to me. I'm sure these are issues that have been discussed in the NSM literature, but as a person who is not knee-deep into the literature, maybe more examples and illustration of the utility would be helpful to convince a skeptic like me that this is actually something useful to do. But again, I admit that maybe my position is biased, so if others are sold on the utility of these explanations, maybe the AC can take this into consideration in making judgments.

The cross-translatability part is interesting and seems like the right way to approach things if quantitative claims about utility are to be made. However, I think BLEU and sentence embedding similarity are weak metrics. I also think without additional human experiments evaluating whether the translation is (1) more comprehensible to the native speakers if the source is NSM, and (2) whether the original lexical meaning really is preserved in the output NSM, are necessary to establish the contribution being claimed. There are some human experiments in the work which is good, but this only concerns rankings among NSMs without taking into consideration fidelity of the NSM explication to the original meaning.

Minor weakness: A lot of the critical metric definitions are pushed to the appendix (e.g., explication score, legality score, circularity...) - they should really be in the main text.

**Questions:**

I described a lot of the conceptual issues I had in the Weaknesses section. These aren't couched in questions, but I'd be happy to engage in discussions about these points.

---

> ### Author Response · Authors · 2025-12-02
> **Response to Reviewers Critiques Towards Paper**
>
> We thank the reviewer for taking the time to write a thoughtful review. We are splitting our response into two parts; in this part, we address specific critiques of our paper and methodology. In the second part, we respond to the reviewer’s conceptual issues with the NSM theory itself, which were emphasized.
>
> ## “Overall framing issues and lack of citations to work being criticized”
>
> We have revised Section 1 to address the framing issues mentioned. For instance, “semantics” is narrowed to “lexical semantics.” We also agree with the reviewer that calling dictionary definitions a “conventional semantic approach” is not well-founded; trained semanticists do not treat dictionaries as a semantic framework. We therefore revised the text to instead contrast our method with “everyday approaches to describing word meanings, such as dictionary-style definitions.” We then added multiple sources to support our claims about the circularity/obscurity/jargon of such approaches:
>
> - Wierzbicka, Anna. "Cups and mugs: Lexicography and conceptual analysis." Australian Journal of Linguistics 4.2 (1984): 205-255.
> - Adamska-Sałaciak, Arleta. "Dictionary definitions: Problems and solutions." Studia Linguistica Universitatis Iagellonicae Cracoviensis 129.4 (2015).
>
> We also appreciate the reviewer bringing to our attention issues with Figure 1. The table of NSM primes is taken from nsm-approach.net/resources – not Wikipedia; but a reference is nonetheless missing and has been added.
>
> ## “The validity of NSM as a theory of meaning”
>
> This perspective was very helpful for us to improve the clarity of Figure 1 and Section 2.1. First, the reviewer makes a sharp observation that DON’T WANT appears like it can be expressed in terms of NOT and WANT. In fact, DON’T and NOT are allolexes of the same prime (see here: https://cdstar.eva.mpg.de/bitstreams/EAEA0-C1BF-E247-4C8F-0/Goddard2011.pdf). Including DON’T WANT in the table is a way of illustrating how WANT and NOT can be packaged together in that grammatical frame, but we agree this ends up being very confusing for exactly the reasons the reviewer identified. We have thus revised the table in Figure 1 to show primes directly next to any allolexes.
> The reviewer also mentions how other NSM primes could be defined in terms of negating each other, giving “LIVE/DIE” as an example. Although this is a reasonable first attempt, the example given here would need a lot more than a simple negation; NOT LIVE can cover an enormous range of states (e.g., never having been alive) that DIE cannot cover. The NSM perspective (and ours, too) is that it should be very, very difficult to accurately paraphrase DIE only in terms of LIVE + other primes. However, the list of NSM primes has changed over time, so changes are not strictly forbidden.
>
> “I also wasn't sure if the NSM explication in in Figure 1b…”
>
> This also revealed many places we could be clearer in the text and Figure 1B. We revised line 115-116 in Section 2.1 to say “The NSM approach is based on the principle that the meaning of any text regardless of its complexity, can be fully paraphrased using only the semantic primes.” Here, “text” may refer either to the target word in isolation, but it can also include the surrounding context that contains it, depending on the scope of the analysis. We agree that Figure 1b is very confusing, as the paraphrase reflects the entire “she was sick,” but the figure makes it appear that only “sick” is being paraphrased, not any of the surrounding context. The figure has been further revised to improve its clarity. We then further edited every figure containing a word/paraphrase example to clearly highlight where the scope of the analysis starts and ends for the given example.

---

> ### Author Response · Authors · 2025-12-02
> **Response to Reviewer’s Conceptual Issues with NSM**
>
> ## “I would like to see a paraphrase of the meanings of words/phrases like..”
>
> _X is blue. [English]_
>
> a. when people think about the colour [m] of X, they can think like this:
>
> b. “the colour [m] of the sky [m] can be like this at many times during the day [m]
>
> c. it can be like this when people can see the sun [m]
>
> d. it can be like this when people can’t see the sun [m]
>
> e. the colour [m] of the sea [m] can be like this”
>
> For more examples of NSM paraphrases for color words, I would direct you to [27], Chapter 4, and [44], page 419. It includes paraphrases for words in different languages that are similar to the English “blue.”
>
> Here is another source that includes a paraphrase for “dogs” (not cats). This paraphrase ends up being many paragraphs long, which makes it impractical to post in its entirety, so we just post the first part.  The full paraphrase starts on paraphrase 549.
> - Cliff, Goddard. "A semantic menagerie: The conceptual semantics of ethnozoological categories." Russian Journal of linguistics 22.3 (2018): 539-559.
>
> a. creatures [m] of one kind
>
> b. they are animals [m], at the same time they are not like animals [m] of other kinds
>
> c. they want to do many things with people, they can do many things as people want
>
> d. many of them live with people because people want this …
>
>
> ## “Maybe more examples and illustration of the utility would be helpful…”
>
> Section H in the Appendix shows several mini-experiments where we applied NSM to downstream tasks such as QA and document retrieval. In the main text, we also cite work demonstrating emerging uses of NSM in various fields—for example, [6] and [38], which show the NSM approach being successfully applied to language teaching/revivalistics. These examples suggest that downstream utility is at least plausible for certain areas.
> That said, any fair evaluation of NSM on downstream tasks would require models that can reliably generate high-quality NSM paraphrases in the first place. We expect the reviewer would agree with this point, even if they remain unsure about the eventual utility of the NSM approach itself. This is exactly the gap our paper is meant to address, and it now enables ongoing work applying these models to downstream tasks, including those mentioned above as well as areas like translation.

---

### Official Review · Reviewer_bejZ · 2025-11-03

**Soundness:** 2
**Presentation:** 3
**Contribution:** 3
**Rating:** 4
**Confidence:** 4

**Summary:**

This paper proposes using LLMs to generate Natural Semantic Metalanguage (NSM) explications—paraphrases of word meanings using a universal set of ~65 semantic primes that exist across all languages. The authors introduce automatic evaluation metrics (legality, substitutability) to assess explication quality and contribute a new dataset for this task. Using this dataset, they fine-tune 1B and 8B parameter models that outperform GPT-4o on explication quality.

**Strengths:**

- This paper addresses an interesting yet underexplored research topic NSM explications, which is relevant to multiple fields including machine translation and instruction following.

- The cross-translatability experiment provides appropriate validation for the effectiveness of fine-tuning.

- The methodologies for the dataset collection and cross-translatability test are well-explained, and the dataset will be a valuable resource for the research community.

**Weaknesses:**

- A limited number of models are applied into the experiments (small size Llama, Gemini-2.0-Flash and GPT-4o), therefore I'm a bit concerned whether the conclusions can be generalized.
- In the substitutability test, the authors use three numbers: $\triangle_{base}$,  $\triangle_{min}$,  $\triangle_{ent}$. All these three are fragile to me (see questions).
- Some of the prompts and examples are in the appendix but not mentioned in main text, which caused difficulty understanding the paragraphs.

**Questions:**

- For $\triangle_{base}$, metric may be sensitive to sample quality. For instance, if a passage provides very clear context for the masked word, the model can predict it correctly without the explication, causing even helpful explications to show marginal improvement in $\triangle_{base}$. Conversely, if the passage is ambiguous, even poor explications might appear helpful.
- For $\triangle_{min}$ and $\triangle_{ent}$, the authors remove 2 sentences from the end of the explication or passage. However, would these metrics remain valid if the redundant information appears at the beginning (or first half) of the explication instead?
- Figure 11 shows the prompt used for generating examples in the substitutability test. However, the prompt does not appear to include explicit controls for example ambiguity. How do the authors assess example quality and ensure appropriate difficulty levels for the substitutability test? This relates to the first concern about sample quality affecting metric reliability.
- It would be better to see some ablation study for the experiments, for example, for the fine-tunes model, what certain ability is improved.

---

> ### Author Response · Authors · 2025-12-02
>
> We thank the reviewer for taking the time to review our paper.
>
> ## “A limited number of models are applied into the experiments… therefore I'm a bit concerned whether the conclusions can be generalized.
>
> We are unsure which conclusions the reviewer believes fail to generalize. The only claims we make about model comparison are:
> - (Line 99) We introduce DeepNSM, two fine-tuned LLMs (1B and 8B), and show that they outperform general LLMs such as GPT-4o and Gemini in explication quality (Section 5).
> - (Line 410) These findings demonstrate that high-quality NSM explication generation is achievable in smaller models with our proposed dataset, addressing the model limitations discussed in Section 2.4 without requiring large-scale compute or commercial APIs.
>
> We do not claim that DeepNSM outperforms all existing LLMs, nor that any arbitrary 8B model (Llama, Gemma, etc.) would achieve the same experimental results. Our conclusions are limited to the models actually evaluated.
>
> ## “Figure 11 shows the prompt used for generating examples in the substitutability test…”
>
> Thank you for pointing this out. In the previous version, the figure labeled as Figure 11 was mistakenly a duplicate of Figure 10. We have corrected this. After reorganizing the appendix, the figure referenced in the comment now appears as Figure 10.
>
> ## “For $\Delta_\text{base}$, metric may be sensitive to sample quality…”
>
> For the first case, this is the reason we use $\Delta_\text{ent}$; by stripping out nearby sentences that may be giving too much context, we reduce the chance that the model can fill in the blank without relying on the explication. This forces the prediction to depend more on the explication itself.
> For the second case, this is why we calculate the delta using the target word’s logits. Even if the passage is ambiguous and the baseline model is uncertain, a good explication should still push probability mass toward the correct word more than a weak or irrelevant one.
>
> ## “For $\Delta_\text{min}$ and $\Delta_\text{ent}$, the authors remove 2…”
>
> For $\Delta_\text{ent}$, the removed sentences aren’t always the final two. The rule is to drop the sentences excluding the one with the target word. So if the passage has three sentences and the target word is in the middle one, then the first and last sentences get removed. If the last sentence has the target word, the previous two get removed. The goal is simply to eliminate any nearby context that might let the model guess the word without relying on the explication.
> For $\Delta_\text{min}$ , the metric assumes that redundant or overly leading information is more likely to appear near the end of an explication rather than at the beginning. That assumption may not be true in all cases, but. Future work could test alternatives—such as removing random sentences instead of the last 2 (or n sentences)—to better measure the redundancy in the explication.
>
> ## “It would be better to see some ablation study for the experiments…”
>
> Table 1 already provides an ablation on the dataset-quality filtering. We train two models; one on filtered data with Explication Score >= 35, and one on randomly selected data with no quality filtering applied. The paragraph starting on line 413 also explains in detail how the filtering affects the fine-tuned model’s behavior, leading to improvements in substitutability and prime usage. Additionally, our response to Reviewer JpBr includes further breakdown on the sensitivity to grader model selection.

---

### Author Response · Authors · 2025-12-02

We thank the reviewers for taking the time to review our paper. We have uploaded a revised version of the manuscript based on reviewer feedback and requests. All changes are highlighted in green. See responses to individual reviewers for details of the edits made.

---

### Meta-Review · Area_Chair_jSYA · 2026-01-08

**Summary:**

Reviewers agree on the novelty of the idea, which boils down to using LLMs to annotate words or sentences with Natural Semantic Metalanguage (NSM) explications? The latter are purported to encapsulate semantic primitives adequate to describe the meaning of words across languages (the primitives are supposed to be universal). Reviewers had fundamental questions about NSM validity, evaluation methodology, language/model coverage, and human evaluation.  The authors did their best to address the reviewers' concerns; however, they did not convincingly address the important question of why NSM vs other semantic formalisms, even if LLMs could do the task with very high accuracy, what do we stand to gain from this? I would have liked to see baselines where NSM is compared against paraphrasing for example, a downstream application where going through NSM helps with the end task (e.g., machine translation), more evaluation in general. For example, can humans annotate text with NSM explications? Do they agree? We expect them to agree to a certain degree, if we are to let the LLM do the task? I appreciated that you cannot correct all the training data, but your test sample is very small. Moreover, why did you not use naturally occurring sentences/words for your training data, there exist corpora annotated with WordNet senses (e.g., Semcore). Finally, you mention correlation with human judgments, how many humans and how many items were used in your correlation (what is the N size and is it statistically significant?). It would also have been good to analyze mistakes in the LLMs' output. What is a bad/good explication?
\

**Reviewer Concerns:**

The authors rephrased some of their claims to better contextualized their work. They further responded to the criticism that they present experiments with a small number of models by arguing they only claim DeepNSM outperforms the specific models tested (GPT-4o, Gemini), not all LLMs. They addressed criticisms that their metrics are fragile by clarifying assumptions and a typo with a figure. They provided examples of explications and engaged meaningfully with the reviewer who criticised the validity of NSM. They also added ablations and a sensitivity analysis for the substitutability score.

**Reviewer Scores:**

I am not sure the reviewers would have changed their scores.

---

### Decision · Program_Chairs · 2026-01-26

Reject